# Hypoxanthine phosphoribosyl transferase 1 metabolizes temozolomide to activate AMPK for driving chemoresistance of glioblastomas

Jianxing Yin[1,2,17,18], Xiefeng Wang[1,2,18], Xin Ge[2,3,18], Fangshu Ding[2,3], Zhumei Shi[1,2], Zehe Ge[2,3], Guang Huang[4], Ningwei Zhao[5,6], Dongyin Chen[7], Junxia Zhang[1,2], Sameer Agnihotri[8], Yuandong Cao[9], Jing Ji[1,2], Fan Lin[2,10], Qianghu Wang[2,11], Qigang Zhou[12], Xiuxing Wang[2,10,13], Yongping You[1,2] ✉, Zhimin Lu[14,15] ✉ & Xu Qian[1,2,3,16] ✉

Temozolomide (TMZ) is a standard treatment for glioblastoma (GBM) patients. However, TMZ has moderate therapeutic effects due to chemoresistance of GBM cells through less clarified mechanisms. Here, we demonstrate that TMZ-derived 5-aminoimidazole-4-carboxamide (AICA) is converted to AICA ribosyl-5-phosphate (AICAR) in GBM cells. This conversion is catalyzed by hypoxanthine phosphoribosyl transferase 1 (HPRT1), which is highly expressed in human GBMs. As the bona fide activator of AMP-activated protein kinase (AMPK), TMZ-derived AICAR activates AMPK to phosphorylate threonine 52 (T52) of RRM1, the catalytic subunit of ribonucleotide reductase (RNR), leading to RNR activation and increased production of dNTPs to fuel the repairment of TMZ-induced-DNA damage. RRM1 T52A expression, genetic interruption of HPRT1-mediated AICAR production, or administration of 6-mercaptopurine (6-MP), a clinically approved inhibitor of HPRT1, blocks TMZ-induced AMPK activation and sensitizes brain tumor cells to TMZ treatment in mice. In addition, HPRT1 expression levels are positively correlated with poor prognosis in GBM patients who received TMZ treatment. These results uncover a critical bifunctional role of TMZ in GBM treatment that leads to chemoresistance. Our findings underscore the potential of combined administration of clinically available 6-MP to overcome TMZ chemoresistance and improve GBM treatment.

Glioblastoma (GBM), a grade IV glioma, is the most common type of primary malignant brain tumor in adults and is also the most lethal cancer of the central nervous system. Temozolomide (TMZ) is the only chemotherapeutic drug that has been confirmed to improve, albeit modestly, the overall survival of GBM patients. The median survival is still only 12–15 months after a standard treatment course[1,2]. TMZ is a small lipophilic prodrug that undergoes spontaneous hydrolysis to become the active metabolite monomethyl triazene 5-(3-methyltriazen-1-yl)-imidazole-4-carboxamide (MTIC). MTIC further reacts with water to liberate 5-aminoimidazole-4-carboxamide

(AICA) and methyldiazonium cations; the latter delivers methyl groups to purine bases of DNA. Methylated purines, especially O6-methylguanine, result in single- and double-strand DNA breaks and cell cycle arrest, which ultimately leads to tumor cell death[3,4].

Intrinsic and acquired TMZ resistance is a major clinical challenge for GBM treatment. TMZ chemoresistance is largely attributed to O-6-methylguanine-DNA methyltransferase (MGMT)-mediated removal of TMZ-induced DNA methyl adducts[5,6] and activation of DNA damage repair systems, such as mismatch repair[7–9], base excision repair[10,11], non-homologous end joining (NHEJ) and homologous recombination (HR) repair[12–14]. Proper DNA damage repair requires adequate pools of deoxyribonucleoside triphosphates (dNTPs), which are produced from deoxyribonucleoside diphosphates (dNDPs) resulting from the reduction of ribonucleoside diphosphates (NDPs) mediated by ribonucleotide reductase (RNR), the key enzyme in the de novo pathway for dNTP biosynthesis[15]. Intensive studies have focused on the regulation of TMZ-derived methyldiazonium cations. However, whether TMZ-derived AICA plays a role in TMZ chemoresistance remains unclear.

In this work, we demonstrate that TMZ-derived AICA is converted to AICAR by HPRT1. AICAR-mediated AMPK activation phosphorylates and activates RNR to produce dNTPs for TMZ-induced DNA damage repair. HPRT1 depletion or 6-MP treatment sensitizes brain tumors to TMZ treatment.

## Results

### AICAR is a bona fide metabolic product of TMZ-derived AICA

TMZ is metabolized into methyldiazonium cations and AICA in physiological conditions (Supplementary Fig. 1a), and methyldiazonium cations cause DNA damage. To determine whether AICA in cells has any physiological functions, we synthesized [15]N-labeled TMZ (Supplementary Fig. 1b) and treated GBM cells with it. As expected, mass spectrum analyses detected [15]N-AICA (Fig. 1a), whose molecular weight (MW) was +1 to that of unlabeled AICA (Supplementary Fig. 1c). Intriguingly, [15]N-AICAR, which contains a ribosyl-5-phosphate group that differs from AICA (Supplementary Fig. 1d), was also identified

(Fig. 1a). TMZ treatment dose-dependently increased the intracellular levels of AICA (Fig. 1b) and AICAR (Fig. 1c). Moreover, direct addition of AICA in the culture medium greatly enhanced the intracellular AICAR amount in GBM cells (Fig. 1d). These results indicated that AICAR is a bona fide intracellular metabolite product of TMZ-derived AICA.

### TMZ-derived AICA activates AMPK

AICAR is an analog of AMP and stimulates AMPK activity, which is critical for many instrumental cellular activities[16,17]. Treatment of GBM cells with TMZ-induced activation of AMPK in a dose- and time-dependent manner, as reflected by elevated phosphorylation of AMPK at T172 and AMPK substrate acetyl-coenzyme A carboxylase 1 (ACC1) at S79 (Fig. 2a and Supplementary Fig. 2a). Consistent with the finding that AICA is a precursor of AICAR, treatment of GBM cells with AICA also induced AMPK activation in a dose- and time-dependent manner (Fig. 2b, c).

We next determined the effects of TMZ-derived methyldiazonium cations on AMPK activation. 2-mercaptoethane sulfonate (Mesna) and WR-1065 were able to effectively react with TMZ-derived methyldiazonium cations both in vitro and in vivo (Supplementary Fig. 2b). In addition, detection of levels of O6-methylguanine (O6-mG), the product of diazonium ion on DNA, by synthesized gold nanoparticles that was reported to specifically recognize O6-mG[18] revealed that pretreating MGMT-null U87 cells with Mesna or WR-1065 effectively reduced O6-mG levels induced by TMZ treatment (Supplementary Fig. 2c), further supporting that Mesna and WR-1065 effectively are able to scavenge TMZ-derived methyldiazonium cations in cells. Consistent with these observations, treatment of GBM cells with Mesna or WR-1065 alleviated TMZ-induced DNA damage as reflected by reduced γ-H2AX (Supplementary Fig. 2d, e). However, Mesna and WR-1065 only exhibited moderate effects on TMZ-induced AMPK activation (Supplementary Fig. 2f, g). Methyldiazonium cations induce the production of reactive oxygen species (ROS)[19], which are believed to activate AMPK[20]. Eliminating ROS by the scavenger N-acetyl-L-cysteine (NAC) had a limited effect on suppressing TMZ-induced AMPK activation (Supplementary Fig. 2h). These results suggested that

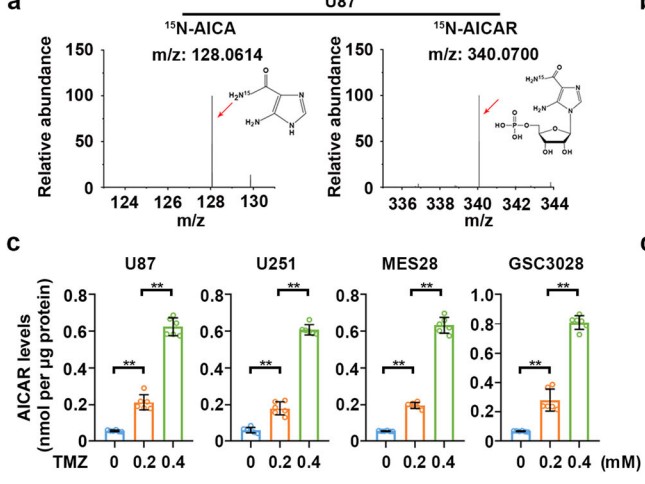

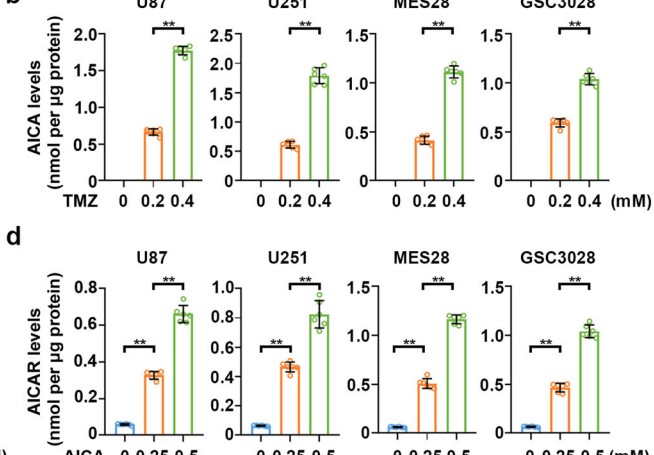

**Fig. 1 | AICAR derived from AICA is a bona fide metabolic product of TMZ.** **a** Representative tandem mass spectra of [15]N-AICA and [15]N-AICAR. **b** Cells were treated with the indicated dose of TMZ for 2 h. Intracellular AICA levels were measured by HPLC-MS. Data represent the mean ± SD from sextuplicate experiments. **P < 0.001. U87, 0.2 vs. 0.4 mM, P = 4.02e-12; U251, 0.2 vs. 0.4 mM, P = 2.5e-09; MES28, 0.2 vs. 0.4 mM, P = 4.56e-10; GSC3028, 0.2 vs. 0.4 mM, P = 2.86e-08. **c** Cells were treated with the indicated dose of TMZ for 2 h. Intracellular AICAR levels were measured by HPLC-MS. Data represent the mean ± SD from sextuplicate experiments. **P < 0.001. U87, 0 vs. 0.2 mM, P = 3.51e-06; 0.2 vs. 0.4 mM, P = 2.16e-08; U251, 0 vs. 0.2 mM, P = 1.93e-05; 0.2 vs. 0.4 mM, P = 5.2e-10;

MES28, 0 vs. 0.2 mM, P = 1.56e-09; 0.2 vs. 0.4 mM, P = 5.18e-10; GSC3028, 0 vs. 0.2 mM, P = 3.93e-05; 0.2 vs. 0.4 mM, P = 4.38e-08. **d** Cells were treated with the indicated dose of AICA for 2 h. Intracellular AICAR was measured by HPLC-MS. Data represent the mean ± SD from sextuplicate experiments. **P < 0.001. U87, 0 vs. 0.25 mM, P = 2.77e-11; 0.25 vs. 0.5 mM, P = 1.79e-08; U251, 0 vs. 0.25 mM, P = 5.11e-11; 0.25 vs. 0.5 mM, P = 4.5e-06; MES28, 0 vs. 0.25 mM, P = 9.92e-10; 0.25 vs. 0.5 mM, P = 3.95e-10; GSC3028, 0 vs. 0.25 mM, P = 7.52e-10; 0.25 vs. 0.5 mM, P = 6.09e-09. Statistics: **b**–**d** unpaired Student's t-test for two-group comparison. Source data are provided as a Source Data file.

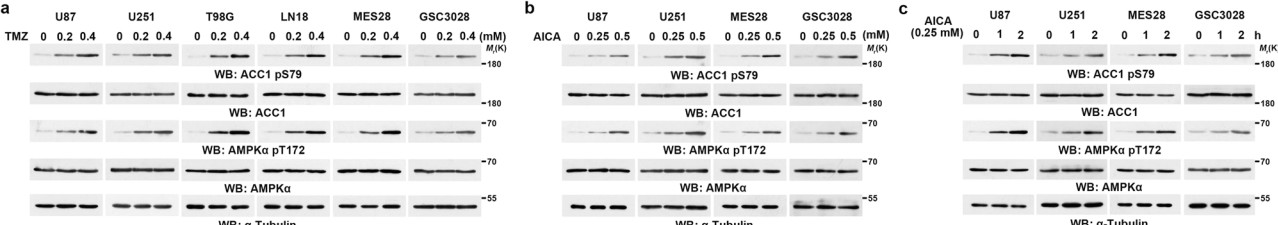

**Fig. 2 | TMZ-derived AICA activates AMPK.** Immunoblot analyses were performed with the indicated antibodies. Three biological repeats were repeated independently with similar results. **a** Cells were treated with the indicated dose of TMZ for

2 h. **b** Cells were treated with the indicated dose of AICA for 2 h. **c** Cells were treated with 0.25 mM of AICA for the indicated time course. Source data are provided as a Source Data file.

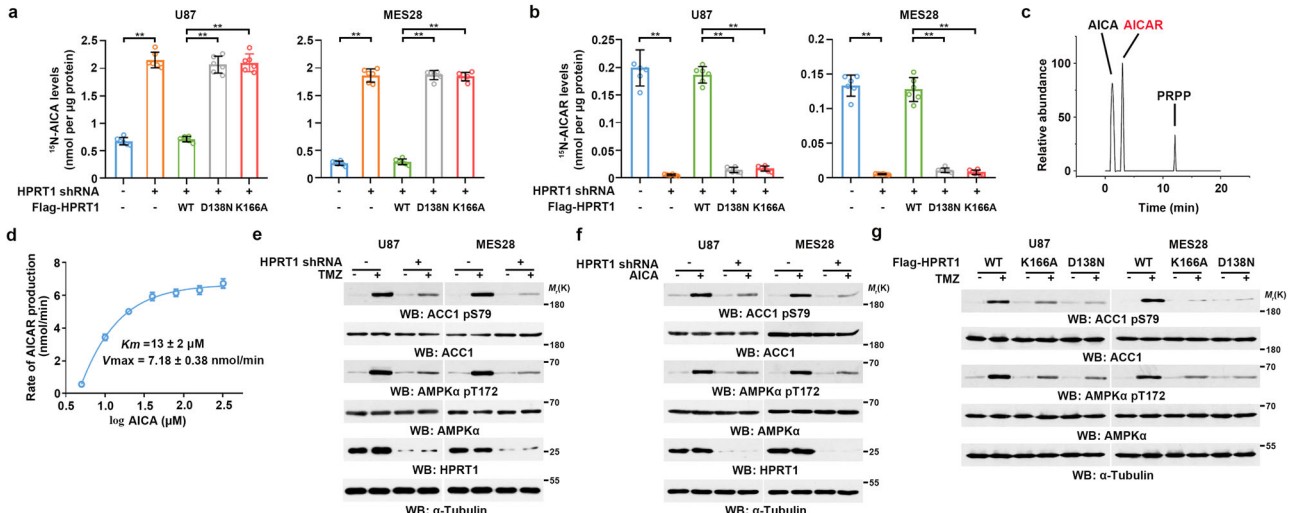

**Fig. 3 | HPRT1 converts TMZ-derived AICA to AICAR. a** Cells were treated with 0.2 mM of $^{15}$N-TMZ for 2 h. Intracellular $^{15}$N-AICA were measured by HPLC-MS. Data represent the mean ± SD from sextuplicate experiments. **$P < 0.001$. U87, Control vs. HPRT1 shRNA, $P = 5.27e-10$; WT vs. D138N, $P = 1.63e-09$; WT vs. K166A, $P = 1.95e-09$; MES28, Control vs. HPRT1 shRNA, $P = 2.98e-11$; WT vs. D138N, $P = 2.61e-12$; WT vs. K166A, $P = 1.96e-12$. **b** Cells were treated with 0.2 mM of $^{15}$N-TMZ for 2 h. Intracellular $^{15}$N-AICAR levels were measured by HPLC-MS. Data represent the mean ± SD from sextuplicate experiments. **$P < 0.001$. U87, Control vs. HPRT1 shRNA, $P = 4.75e-08$; WT vs. D138N, $P = 1.18e-10$; WT vs. K166A, $P = 1.32e-10$; MES28, Control vs. HPRT1 shRNA, $P = 1.61e-09$; WT vs. D138N, $P = 1.67e-08$; WT vs. K166A, $P = 1.31e-08$. **c** Representative chromatograms of products of the HPRT1 kinase assay. **d** Michaelis-Menten curve of HPRT1 for AICA. Reactions were

performed by mixing purified active HPRT1 and AICA. Data represent the mean ± SD from sextuplicate experiments. **e** and **f** Cells with or without shRNA-mediated HPRT1 depletion were treated with or without 0.2 mM of TMZ (**e**) or 0.25 of mM AICA for 2 h (**f**), respectively. Immunoblot analyses were performed with the indicated antibodies. Three biological repeats were repeated independently with similar results. **g** HPRT1-depleted cells with reconstituted expression of WT Flag-HPRT1, Flag-HPRT1 D138N, or Flag-HPRT1 K166A were treated with or without 0.2 mM of TMZ for 2 h. Immunoblot analyses were performed with the indicated antibodies. Three biological repeats were repeated independently with similar results. Statistics: **a**, **b** unpaired Student's *t*-test for two-group comparison. Source data are provided as a Source Data file.

TMZ induces activation of AMPK primarily through the production of AICA and the subsequent AICAR instead of methyldiazonium cations.

## HPRT1 converts TMZ-derived AICA to AICAR

Given that AICAR contains a ribosyl-5-phosphate group that differs from AICA (Supplementary Fig. 1d), we speculated that AICA is converted to AICAR by a phosphoribosyl transferase-mediated reaction. To test this hypothesis, we depleted the five known enzymes that are responsible for these reactions, hypoxanthine phosphoribosyl transferase 1 (HPRT1), adenine phosphoribosyl transferase (APRT), orotate phosphoribosyl transferase (OPRT/UMPS), uracil phosphoribosyl transferase (UPRT), and quinolinate phosphoribosyl transferase (QPRT), in U87 and MES28 GBM cells. Depletion of HPRT1, but not the other four enzymes, resulted in increased accumulation of $^{15}$N-AICA (Fig. 3a, and Supplementary Fig. 3a) and decreased $^{15}$N-AICAR production (Fig. 3b and Supplementary Fig. 3b); these effects were reversed by reconstituted expression of RNA interference-resistant (r) wild-type (WT) HPRT1, but not by catalytically inactive rHPRT1 D138N or K166A mutant[21–23] (Fig. 3a, b and Supplementary Fig. 3c).

These results suggested that HPRT1, whose primary role is to convert hypoxanthine into inosine monophosphate (IMP) and guanine into guanosine monophosphate (GMP)[24], catalyzes AICA to AICAR.

To support this finding, we performed an in vitro reaction by mixing AICA with bacterially purified HPRT1 in the presence of phosphoribosyl pyrophosphate (PRPP) as the donor of ribosyl-5-phosphate. Purified WT HPRT1, but not catalytically inactive HPRT1 D138N or HPRT1 K166A (Supplementary Fig. 3d, e), converted AICA to AICAR (Fig. 3c) with Michaelis-Menten constant ($Km$) at 12.92 ± 2.48 μM and Vmax at 7.18 ± 0.38 nmol/min (Fig. 3d). Although the Vmax values were comparable, the $Km$ values were much higher than those of physiological HPRT1 substrates, such as hypoxanthine ($Km = 1.45 ± 0.26$ μM, Vmax = 8.81 ± 0.18 nmol/min) and guanine ($Km = 2.25 ± 0.31$ μM, Vmax = 9.51 ± 0.31 nmol/min) (Supplementary Fig. 3f), indicating that AICA is a poor substrate for HPRT1. However, the intracellular concentration of AICA was much greater than the $Km$ of HPRT1 for AICA even at low amounts (>50 μM) of TMZ treatment (Supplementary Fig. 3g), compensating the low efficiency of this reaction and suggesting that the HPRT1-mediated AICA conversion is sufficient after

treatment of GBM cells with TMZ. These in vitro and in vivo results indicated that HPRT1 converts AICA to AICAR in response to TMZ treatment.

We next examined the role of HPRT1 in TMZ-induced AMPK activation. Depletion of HPRT1, but not other phosphoribosyl transferases, inhibited both TMZ- (Fig. 3e and Supplementary Fig. 3h) and AICA-induced AMPK activation (Fig. 3f). This inhibition was abrogated by reconstituted expression of WT rHPRT1 but not the rHPRT1 D138N and rHPRT1 K166A mutants (Fig. 3g). These results indicated that HPRT1-mediated conversion of AICA to AICAR is responsible for TMZ-induced AMPK activation.

## TMZ-activated AMPK phosphorylates RRM1 at T52

TMZ treatment causes substantial DNA damage in tumor cells[25], whereas enhanced DNA damage repair mainly contributes to TMZ resistance[11,26]. Pretreatment of GBM cells with AICA or the AMPK activator A769662 reduced TMZ-induced cell apoptosis (Supplementary Fig. 4a, b) with shortened γ-H2AX dynamics (Fig. 4a, b), reflecting the protective effects of AICA and AMPK activation. In contrast, simultaneous deletion of AMPKα1 and AMPKα2 by CRISPR-mediated gene editing enhanced TMZ-induced cell apoptosis (Supplementary Fig. 4c) with prolonged γ-H2AX dynamics (Supplementary Fig. 4d). These results suggested that TMZ-produced AICA and AMPK activation protect GBM cells from TMZ-induced DNA damage.

To determine the mechanism underlying AICA-/AMPK-mediated DNA damage repair, we immunoprecipitated Flag-AMPKα1 from TMZ-treated U87 cells. Mass spectrometry analyses of the immunoprecipitates showed that RRM1, the catalytic subunit of RNR, is an associated protein (Supplementary Data 1). The TMZ-induced association between AMPKα1 and RRM1 was further confirmed by coimmunoprecipitation analyses with an antibody against RRM1 or AMPKα1 (Fig. 4c). Notably, mass spectrometry analyses of RRM1 immunoprecipitates from the TMZ-treated GBM cells showed that RRM1 was phosphorylated at T52 (Supplementary Fig. 4e and Supplementary Table 1), an evolutionarily conserved residue (Supplementary Fig. 4f). Scansite analyses (http://scansite3.mit.edu/) of the RRM1 protein sequence revealed that T52 is a potential phosphorylation residue of AMPK, albeit that this amino acid residue was not perfectly fit to optimal AMPK phosphorylation motif (Supplementary Fig. 4g). To determine whether AMPK phosphorylates RRM1 at T52, we performed in vitro kinase assay by incubating active AMPK, validated by SAMS peptide[27] as a substrate (Supplementary Fig. 4h), with bacterially purified WT His-RRM1 or His-RRM1 T52A phosphorylation-dead mutant (Supplementary Fig. 4i) in the presence of [γ-$^{32}$P]ATP, followed by autoradiography. Figure 4d showed that AMPK phosphorylates WT RRM1, but not RRM1 T52A mutant. This result was further confirmed by a generated specific antibody recognizing RRM1 pT52 (Fig. 4e and Supplementary Fig. 4j). In addition, stoichiometry analyses of RRM1 phosphorylated by AMPK showed incorporation of ~1 mol of phosphate per mol of RRM1 proteins (Fig. 4f). These in vitro observations were recapitulated in GBM cells where TMZ treatment promoted RRM1 phosphorylation at T52; this phosphorylation was abrogated by RRM1 T52A mutation (Fig. 4g). In addition, AMPKα1/2 deficiency abolished TMZ-induced phosphorylation of RRM1 T52 (Fig. 4h). Consistently, HPRT1 depletion, which substantially reduced TMZ-induced AMPK activation, suppressed RRM1 T52 phosphorylation (Fig. 4i). These results indicated that HPRT1-mediated AMPK activation phosphorylates RRM1 at T52 in GBM cells upon TMZ treatment.

## TMZ-mediated RRM1 T52 phosphorylation promotes DNA damage repair

To test whether RRM1 T52 phosphorylation regulates RNR activity, we performed in vitro RNR activity assays by mixing bacterially purified WT His-RRM1 or His-RRM1 T52A with bacterially purified His-RRM2, the regulatory subunit of RNR (Supplementary Fig. 5a), in the presence or

absence of AMPK. AMPK-phosphorylated WT RRM1, but not RRM1 T52A, substantially increased RNR activity (Fig. 5a). RNR activity is controlled by (deoxy, d)ATP:ATP ratio. The binding of dATP inhibits, whereas binding of ATP activates, RNR[15]. The enzyme activity of RNR with CDP as substrates and increasing concentrations of ATP showed that the apparent $K_d$ value of ATP for activating RNR was reduced from 0.37 ± 0.03 mM to 0.02 ± 0.005 mM when RRM1 was phosphorylated at T52 by AMPK; this effect was abolished by RRM1 T52A (Fig. 5b). On the other hand, T52 phosphorylation of RRM1 had no impact on the inhibitory effect of dATP on RNR (Supplementary Fig. 5b). These results suggested that AMPK-phosphorylated RRM1 T52 increases the binding affinity of RNR for ATP, thereby increasing RNR activities.

Adequate dNTPs generated by RNR are essential for efficient DNA damage repair[15,26]. Depletion of endogenous RRM1 and reconstituted expression of Flag-tagged WT RRM1 or RRM1 T52A in GBM cells (Supplementary Fig. 5c) showed that TMZ treatment only increased activity of RNR containing WT RRM1 (Supplementary Fig. 5d). In addition, AMPKα1/2 deficiency (Supplementary Fig. 5e) or HPRT1 depletion (Fig. 5c) abrogated TMZ-increased RNR activity. Notably, HPRT1 deletion or RRM1 T52A expression impeded DNA damage repair, as evidenced by prolonged detection of γ-H2AX (Supplementary Fig. 5f, g), enhanced cell apoptosis (Fig. 5d, e), and reduced cell proliferation (Supplementary Fig. 5h, i) upon TMZ treatment. In addition, HPRT1 expression levels in different GBM cells were positively correlated with the IC50 of TMZ for apoptosis (Supplementary Fig. 5j, k). These results indicated that TMZ-induced and HPRT1-activated AMPK phosphorylates RRM1 at T52 to promote the repair of TMZ-induced DNA damage, therefore facilitating cell survival under TMZ treatment.

The expression status of O6-methylguanine-DNA methyltransferase (MGMT) predicts chemo-responses of TMZ[28]. However, depletion of HPRT1 had no discriminated effects on the repair dynamic of TMZ-induced DNA damage when comparing between MGMT-null U87 and MGMT-intact MES28 cells (Supplementary Figs. 5f and 6a). In consistent with these observations, depletion of MGMT in MGMT-intact LN229 and LN18 cells or overexpression of MGMT in MGMT-null U87 and U251 cells had no additive effects on HPRT1 depletion-induced DNA repair (Supplementary Fig. 6b, c) or apoptosis (Supplementary Fig. 6d, e) in response to TMZ treatment. These results suggested that HPRT1-mediated DNA repair is independent of MGMT status.

These above results revealed a critical role of HPRT1-mediated DNA repair in primary/intrinsic resistance to TMZ. In addition to this, we determined whether HPRT1-mediated DNA repair played a role in secondary/acquired resistance to TMZ. To do so, we employed our previously generated TMZ-resistant cell lines and their paired counterparts, U251T3rd vs. U251S, and N3T3rd vs. N3S[29]. We confirmed the resistant capacity of these cells to TMZ (Supplementary Fig. 7a). HPRT1 expression levels were much increased in these TMZ-resistant cells (Supplementary Fig. 7b). In line with their increased HPRT1 expression levels, the TMZ-resistant cells exhibited much accelerated AMPK activation and the subsequent phosphorylation of RRM1 T52 in response to both dosage- and time-dependent TMZ treatment (Supplementary Fig. 7c, d). Depletion of HPRT1 in TMZ-resistant cells suppressed TMZ-induced AMPK activation and phosphorylation of RRM T52 (Supplementary Fig. 7e), and subsequently increased the sensitivity to TMZ treatment as demonstrated by the reduced IC50 to TMZ (Supplementary Fig. 7f) and increased apoptotic effects (Supplementary Fig. 7g). These data suggested that HPRT1-mediated AMPK/RRM1/DNA repair pathway played a role in secondary/acquired resistance to TMZ.

## Inhibition of HPRT1-mediated RRM1 T52 phosphorylation sensitizes brain tumors to TMZ treatment

To determine the role of HPRT1 in brain tumor resistance to TMZ treatment in animals, we intracranially injected nude mice with

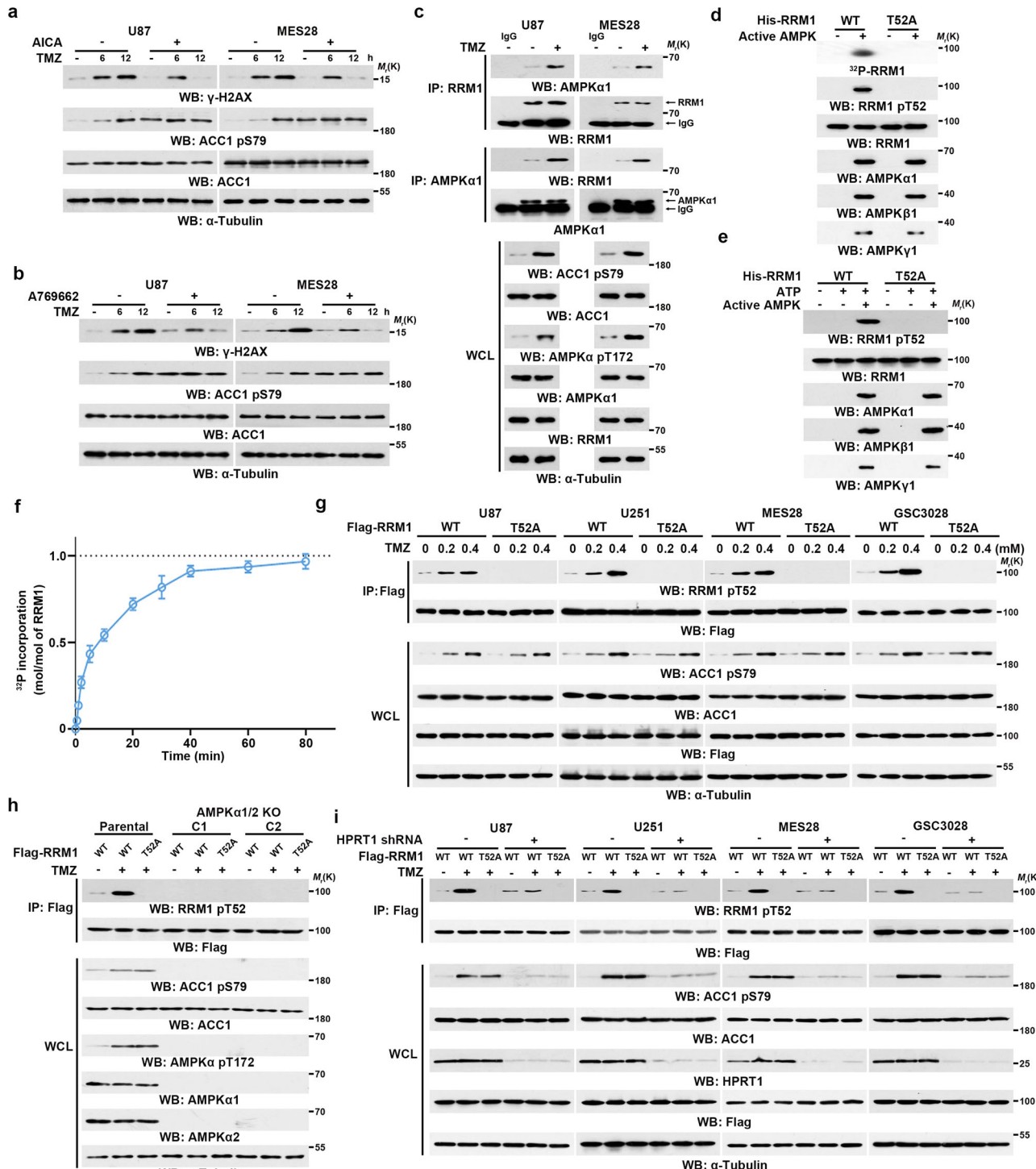

**Fig. 4 | TMZ-activated AMPK phosphorylates RRM1 at T52. a–e, g–i** Immunoprecipitation and immunoblot analyses were performed with the indicated antibodies. **a–i** Three biological repeats were repeated independently with similar results. **a** Cells pretreated with or without 0.25 mM of AICA were treated with or without 0.2 mM of TMZ for the indicated time points. **b** Cells pretreated with or without 0.25 mM of A769662 were treated with or without 0.2 mM of TMZ for the indicated time points. **c** Cells were treated with or without 0.2 mM of TMZ for 2 h. **d** In vitro phosphorylation and SDS-PAGE analysis and autoradiography were performed by mixing purified WT His-RRM1 or His-RRM1 T52A protein with active AMPK in the presence of [γ-$^{32}$P]ATP. **e** Bacterially purified WT His-RRM1 or His-RRM1 T52A was incubated with or without active AMPK in the

presence or absence of ATP. **f** Stoichiometry of RRM1 phosphorylation by AMPK. Bacterially purified His-RRM1 was incubated with active AMPK in the presence of [γ-$^{32}$P]ATP. The radioactive intensity of incorporated $^{32}$P was measured and the incorporation of $^{32}$P into RRM1 was calculated. Data represent the mean ± SD of triplicate samples. **g** The indicated cells expressing WT Flag-RRM1 or Flag-RRM1 T52A were treated with the indicated dose of TMZ for 2 h. **h** AMPKα1/2 double knockout (DKO) U87 cells expressing WT Flag-RRM1 or Flag-RRM1 T52A were treated with or without 0.2 mM of TMZ for 2 h. C1 and C2, two clones of AMPKα1/2 DKO U87 cells. **i** The indicated cells with or without HPRT1 depletion were transfected with vectors expressing WT Flag-RRM1 or Flag-RRM1 T52A. The cells were further treated with 0.2 mM of TMZ for 2 h.

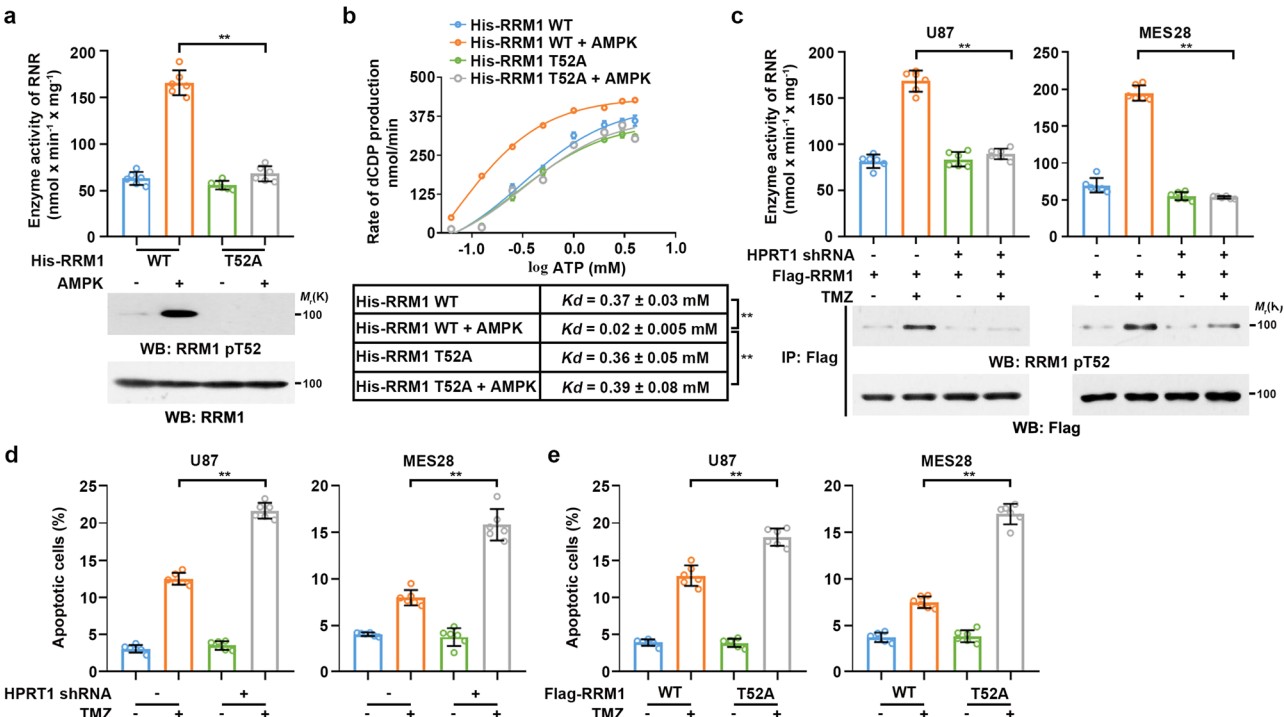

**Fig. 5 | TMZ-mediated RRM1 T52 phosphorylation promotes DNA damage repair. a** and **b** WT His-RRM1 or His-RRM1 T52A mutant proteins were mixed with His-RRM2 and incubated with active AMPK in the presence of ATP for 20 min followed by HPLC-MS analysis. (**a**) Ribonucleotide reductase (RNR) activity was measured according to dCDP production. (**b**) $Kd$ was calculated. Data represent the mean ± SD from sextuplicate experiments. **$P < 0.001$. **a** WT + AMPK vs. T52A + AMPK, $P = 2.91e-08$; **b** His-RRM1 WT vs. His-RRM1 WT + AMPK, $P = 3.73e-05$; His-RRM1 WT + AMPK vs. His-RRM1 T52A + AMPK, $P = 0.00032$. **c** Cells with or without HPRT1 depletion were transfected with Flag-RRM1 and treated with or without 0.2 mM of TMZ for 2 h. RNR activity was measured according to dCDP production. Immunoblotting analysis was performed to confirm the AMPK-mediated phosphorylation status of RRM1. Data represent the mean ± SD from sextuplicate experiments. **$P < 0.001$. U87, Flag-RRM1 + TMZ vs. Flag-

RRM1 + TMZ + HPRT1 shRNA, $P = 3.18e-08$; MES28, Flag-RRM1 + TMZ vs. Flag-RRM1 + TMZ + HPRT1 shRNA, $P = 1.56e-11$. **d** Cells with or without HPRT1 depletion were treated with or without 0.2 mM of TMZ, respectively, for 24 h. The rate of apoptotic cells was examined by FACS. Data represent the mean ± SD from sextuplicate experiments. **$P < 0.001$. U87, TMZ vs. HPRT1 shRNA + TMZ, $P = 1.12e-08$; MES28, TMZ vs. HPRT1 shRNA + TMZ, $P = 1.29e-06$. **e** RRM1-depleted cells with reconstituted expression of Flag-RRM1 WT or Flag-RRM1 T52A were treated with or without 0.2 mM of TMZ, respectively, for 24 h. The rate of apoptotic cells was examined by FACS. Data represent the mean ± SD from sextuplicate experiments. **$P < 0.001$. U87, Flag-RRM1 WT + TMZ vs. Flag-RRM1 T52A + TMZ, $P = 3.68e-05$; MES28, Flag-RRM1 WT + TMZ vs. Flag-RRM1 T52A + TMZ, $P = 4.7e-09$. Statistics: **a**–**e** unpaired Student's *t*-test for two-group comparison. Source data are provided as a Source Data file.

luciferase-expressing MES28 or U87 cells with or without HPRT1 depletion. After tumor cell implantation, mice were treated with TMZ (20 mg/kg, i.p., for 5 consecutive days) or the control vehicle, followed by bioluminescent monitoring of tumor growth using an In Vivo Imaging System (IVIS) (Supplementary Fig. 8a). Depletion of HPRT1 reduced tumor growth and substantially sensitized the brain tumors to TMZ treatment (Fig. 6a and Supplementary Fig. 8b, c). Intriguingly, combining HPRT1 depletion and TMZ treatment achieved the longest survival extension (Fig. 6b and Supplementary Fig. 8d). HPRT1 depletion (Supplementary Fig. 8e, f) inhibited TMZ treatment-induced AMPK activation in brain tumor tissues (Fig. 6c and Supplementary Fig. 8g) and enhanced TMZ-induced γ-H2AX levels (Fig. 6d and Supplementary Fig. 8h) and apoptosis (Fig. 5e and Supplementary Fig. 8i).

To further support the finding that HPRT1-regulated RRM1 activation plays a role in TMZ chemoresistance, we intracranially injected nude mice with luciferase-expressing MES28 or U87 cells with depletion of RRM1 and reconstituted expression of the WT Flag-RRM1 or Flag-RRM1 T52A mutant. The expression of RRM1 T52A showed slight effects on tumor growth (Fig. 6f and Supplementary Fig. 9a, b), which is consistent with the observations that the T52A mutation had limited effects on RNR activity in the absence of AMPK activation (Fig. 5a). In sharp contrast, RRM1 T52A expression strongly enhanced the inhibitory effects of TMZ on brain tumors (Fig. 6f and Supplementary Fig. 9a, b) with corresponding extension of survival

time (Fig. 6g). Moreover, combining RRM1 T52A expression and TMZ treatment achieved the longest survival extension (Supplementary Fig. 9c). IHC results indicated that RRM1 T52A expression (Supplementary Fig. 9d, e) was accompanied with increased γ-H2AX levels (Fig. 6h and Supplementary Fig. 9f) and apoptosis (Fig. 5i and Supplementary Fig. 9g). Together, these results strongly suggested that HPRT1-mediated AMPK activation and subsequent RNR activation are instrumental for brain tumor chemoresistance to TMZ treatment.

**Combined treatment with 6-MP and TMZ blocks TMZ-induced DNA damage repair and synergistically inhibits brain tumor growth**

We next explored the currently available therapeutic approaches to intervene in TMZ chemoresistance. The thiopurine drug 6-mercaptopurine (6-MP), which has been used for acute lymphoblastic leukemia (ALL) and chronic myeloid leukemia (CML) treatment, competes with HPRT1 substrates and thereby inhibits HPRT1 activity[30]. Consistent with the previous report[31], the $Km$ of HPRT1 for 6-MP was $44 ± 6$ nM (Supplementary Fig. 10a), which was considerably lower than that for AICA ($13 ± 2$ μM, Fig. 3d), suggesting that 6-MP is able to competitively prevent AICA binding to HPRT1. As expected, 6-MP treatment increased the intracellular accumulation of [15]N-AICA (Fig. 7a) and decreased the production of [15]N-AICAR (Fig. 7b) in the [15]N-TMZ-treated GBM cells. A synergistic effect of combining 6-MP and TMZ

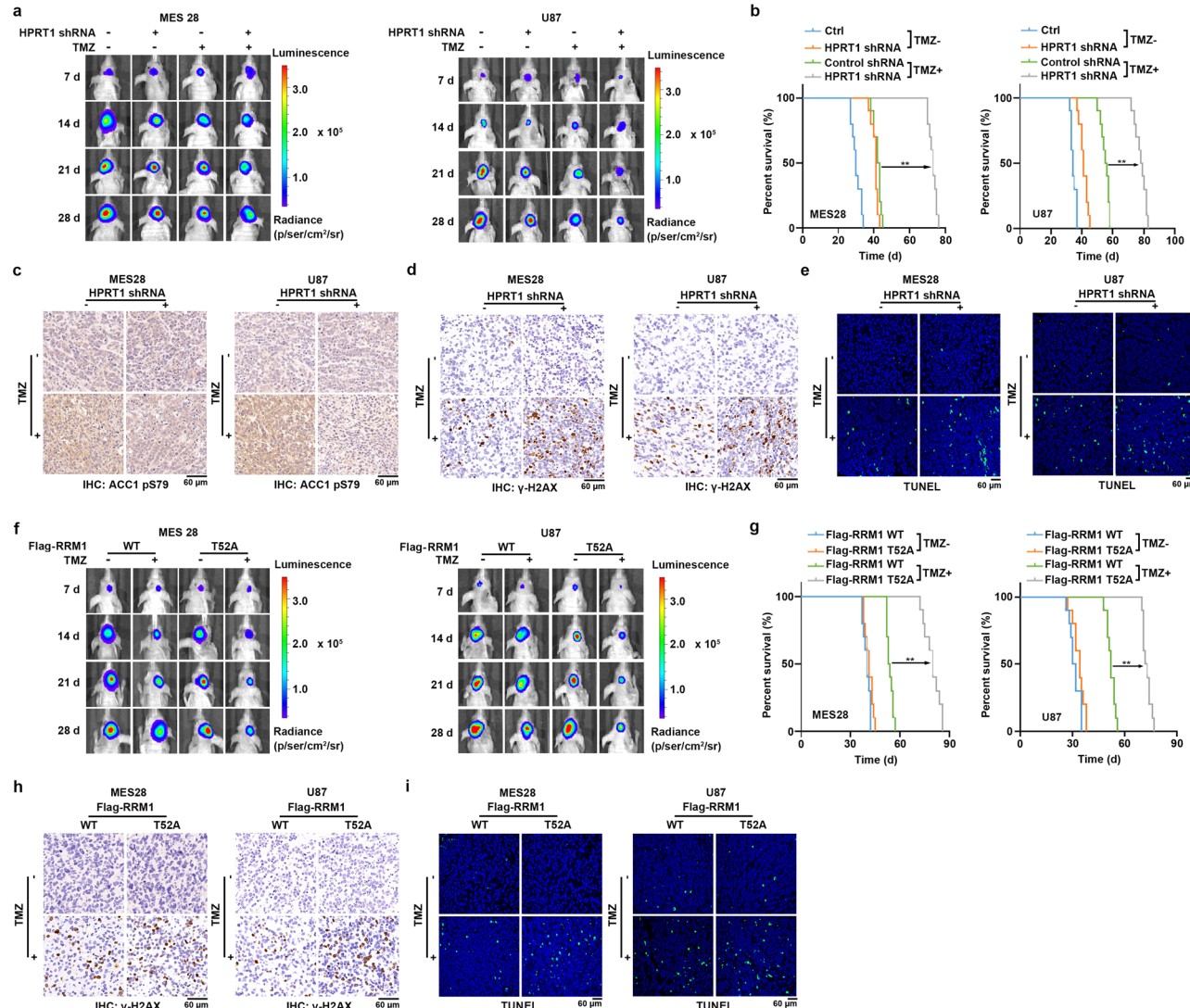

**Fig. 6 | Inhibition of HPRT1-mediated RRM1 T52 phosphorylation sensitizes brain tumors to TMZ treatment. a** Luciferase-expressing MES28 and U87 cells with or without HPRT1 depletion were intracranially injected into nude mice ($n = 10$ for each group). Shown are luminescence intensity of tumors in representative mice at the indicated time points. **b** The survival time of the indicated groups of mice was recorded. **$P < 0.001$. MES28, Control shRNA+TMZ vs. HPRT1 shRNA +TMZ, $P = 1.97e{-}05$; U87, Control shRNA+TMZ vs. HPRT1 shRNA+TMZ, $P = 2.81e{-}05$. **c** and **d** Representative IHC images of ACC1 pS79 (**c**) and γ-H2AX (**d**) were shown. Scale bars, 60 μm. $n = 10$ for each group. **e** Representative TUNEL images were shown. Scale bars, 60 μm. $n = 10$ for each group. **f** Luciferase-expressing MES28 and U87 cells with RRM1 depletion and reconstituted expression of WT Flag-RRM1 or the Flag-RRM1 T52A mutant were intracranially injected into nude mice ($n = 10$ for each group). Shown are luminescence intensity of tumors in representative mice at the indicated time points. **g** The survival time of the indicated groups of mice was recorded. **$P < 0.001$. MES28, Flag-RRM1 WT + TMZ vs. Flag-RRM1 T52A + TMZ, $P = 4.42e{-}05$; U87, Flag-RRM1 WT + TMZ vs. Flag-RRM1 T52A + TMZ, $P = 0.00025$. **h** Representative IHC images of RRM1 pT52 were shown. Scale bars, 60 μm. $n = 10$ for each group. **i** Representative TUNEL images were shown. Scale bars, 60 μm. $n = 10$ for each group. Statistics: **b**, **g** Log-rank test for two-group comparison. **b** Control shRNA + TMZ vs. HPRT1 shRNA + TMZ, **g** Flag-RRM1 WT + TMZ vs. Flag-RRM1 T52A + TMZ. Source data are provided as a Source Data file.

in GBM cells was observed (Fig. 7c). In addition, compared with treatment with TMZ or 6-MP alone, 6-MP treatment substantially blocked TMZ-induced AMPK activation and subsequent AMPK-mediated RRM1 T52 phosphorylation (Supplementary Fig. 10b). Treating cells with 10μM of 6-MP, which was far lower than its IC50 to GBM cells (Supplementary Fig. 10c), significantly increased TMZ sensitivity as reflected by suppressed cell proliferation (Supplementary Fig. 10d) with prolonged expression of γ-H2AX (Supplementary Fig. 10e) and increased apoptosis (Supplementary Fig. 10f).

Next, tumor-bearing mice were treated with TMZ (20 mg/kg, i.p., for 5 consecutive days) in combination with or without 6-MP (20 mg/ kg, i.p., for 5 consecutive days) or the control vehicle (Supplementary Fig. 11a). The hematoxylin and eosin (H&E) staining and IHC staining with cleaved caspase-3 confirmed that 6-MP treatment had no

observed damage on the morphologies or viability of mouse hepatocytes, pneumonocyte, or renal cells (Supplementary Fig. 11b–d), suggesting that 6-MP treatment exerted no severe toxicity to normal tissues. Compared with treatment with TMZ or 6-MP alone, the combination of 6-MP and TMZ substantially inhibited tumor growth (Fig. 7d, and Supplementary Fig. 11e, f). This combined treatment largely extended mouse survival time (Fig. 7e, and Supplementary Fig. 11g). Additionally, this combined treatment also largely inhibited TMZ-induced AMPK activation (Fig. 7f, and Supplementary Fig. 11h), RRM1 pT52 levels (Fig. 7g, Supplementary Fig. 11i) and substantially enhanced the levels of γ-H2AX (Fig. 7h, and Supplementary Fig. 11j) and apoptosis in tumor tissues (Fig. 7i and Supplementary Fig. 11k). In consistent with these observations, measuring O6-mG levels by binding intensity of MGMT C145S mutant with O6-mG (Supplementary Fig. 11l) revealed that

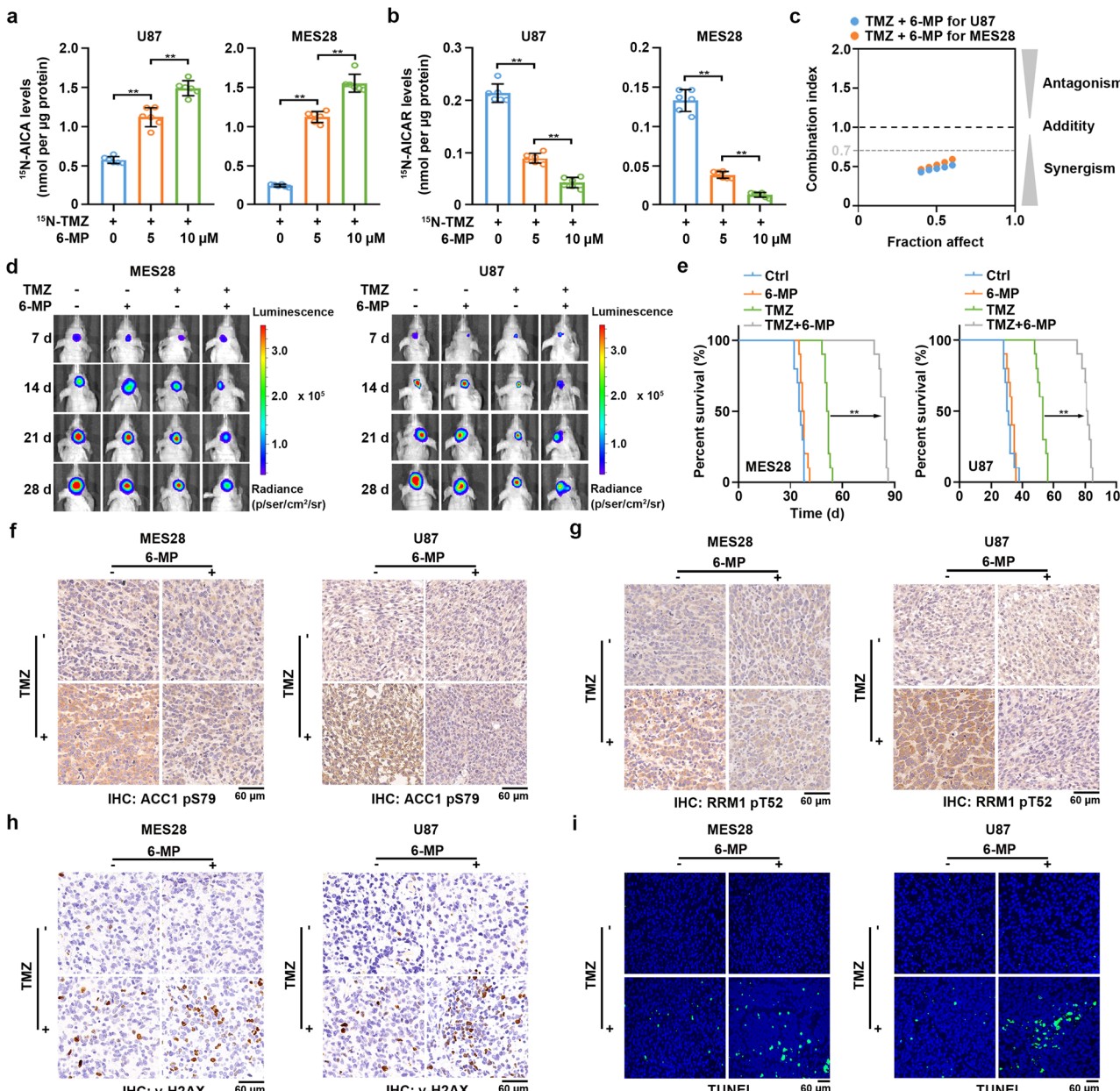

**Fig. 7 | Combined treatment with 6-MP and TMZ blocks TMZ-induced DNA damage repair and synergistically inhibits brain tumor growth. a** and **b** Cells were treated with 0.2 mM of $^{15}$N-TMZ together with the indicated concentration of 6-MP for 2 h. Intracellular $^{15}$N-AICA (**a**) and $^{15}$N-AICAR levels (**b**) were measured by HPLC-MS. Data represent the mean ± SD from sextuplicate experiments. **P < 0.001. **c** Synergistic effect of TMZ with 6-MP on U87 (gray circle), and MES28 (orange circle). CI (combination index) value was calculated. **d** Luciferase-expressing MES28 and U87 cells were intracranially injected into nude mice (n = 10

for each group). Shown are luminescence intensity of tumors in representative mice at the indicated time points. **e** The survival time of the indicated groups of mice was recorded. **P < 0.001. **f**–**h**, Representative IHC images of ACC1 pS79 (**f**), RRM1 pT52 (**g**), and γ-H2AX (**h**) were shown. Scale bars, 60 μm. n = 10 for each group. **i** Representative TUNEL images were shown. Scale bars, 60 μm. n = 10 for each group. Statistics: **a**, **b** unpaired Student's t-test for two-group comparison. **e** Log-rank test for two-group comparison (TMZ vs. TMZ + 6-MP). Source data are provided as a Source Data file.

6-MP and TMZ combination greatly increased O6-mG levels in tumor tissues when comparing with TMZ treatment alone (Supplementary Fig. 11m). In contrast, no additive effects of 6-MP and TMZ combination on the levels of O6-mG than TMZ treatment alone were observed in the tumor-adjacent or normal brain tissues (Supplementary Fig. 11n, o), probably due to the expression of MGMT which effectively removes O6-mG in normal brain tissues, therefore protecting normal brain tissues from damaging by 6-MP and TMZ combination treatment. Collectively, these results indicated that combined treatment with 6-MP and TMZ blocks TMZ-induced DNA damage repair and synergistically inhibits brain tumor growth.

## HPRT1 expression predicts poor prognosis of brain tumor patients

Analysis of the TCGA database revealed that HPRT1 expression was higher in GBMs than in low-grade gliomas (Supplementary Fig. 12a). To determine the clinical relevance of HPRT1 to GBM progression, we performed IHC staining of HPRT1 in our collected 100 primary GBM samples from patients who received surgeries prior to standard care including TMZ treatment (Supplementary Fig. 12b), and evaluated the IHC score of HPRT1 expression (Supplementary Fig. 12c and Supplementary Data 2). HPRT1 expression levels were inversely correlated with overall survival of GBM patients (Fig. 8a, P = 0.0025). This

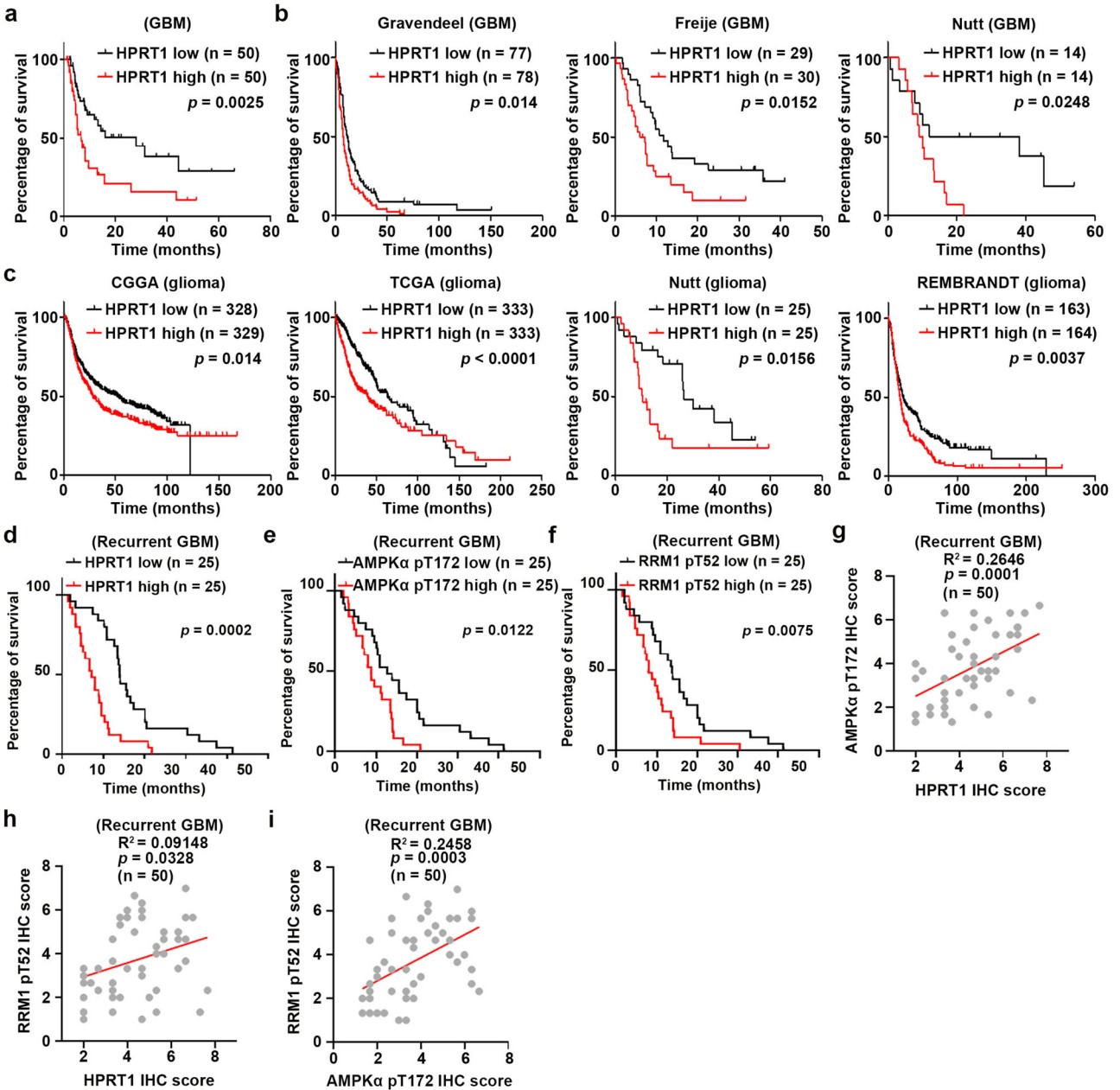

**Fig. 8 | HPRT1 expression predicts poor prognosis of GBM patients. a** and **b** Kaplan–Meier survival analysis based on HPRT1 expression from collected primary GBM samples (**a**) and the indicated GBM datasets (**b**). **c** Kaplan–Meier survival analysis based on HPRT1 expression from the indicated glioma datasets. **d**–**f** Kaplan–Meier survival analysis based on HPRT1 (**d**), AMPK pT172 (**e**), and RRM1 pT52 (**f**) expression from collected recurrent GBM samples. **g**–**i** The Pearson correlation test was used to analyze the correlation among HPRT1, AMPK pT172, and RRM1 pT52. Statistics: **a**–**f** Log-rank test for two-group comparison. Source data are provided as a Source Data file.

negative relationship between HPRT1 expression and GBM patient survival was also reproduced in different datasets, including the Gravendeel ($P = 0.014$)[32], Freije ($P = 0.0152$)[33], Nutt ($P = 0.0248$)[34] (Fig. 8b), Joo ($P = 0.4208$)[35], and Phillips ($P = 0.2622$)[36] (Supplementary Fig. 12d). In addition, similar results were obtained in glioma patients from additional datasets, including the CGGA ($P = 0.014$)[37], TCGA ($P < 0.0001$)[38], Nutt ($P = 0.0156$), Rembrandt ($P = 0.0037$)[39] (Fig. 8c), POLA ($P = 0.0152$)[40], Freije ($P = 0.0627$), Phillips ($P = 0.4548$), and Kamoun ($P = 0.1164$)[41] (Supplementary Fig. 12e). These results demonstrated that HPRT1 predicts poor prognosis of primary gliomas.

We next validated the HPRT1/AMPK/RRM1 signal cascade in primary GBM samples. In our collected 100 primary GBM samples, the levels of HPRT1 expression showed no correlation with AMPK pT172 or

RRM1 pT52 (Supplementary Fig. 12f, g). The levels of AMPK pT172 demonstrated positive correlation with that of RRM1 pT52 in all these samples ($n = 100$, Supplementary Fig. 12h), as well as in those samples expressing with low or high levels of HPRT1 ($n = 50$, respectively, Supplementary Fig. 12i). In addition, no obvious correlation was observed between levels of AMPK pT172 or RRM1 pT52 with survival of the patients (Supplementary Fig. 12j, k). Analyzing information of AMPK pT172 levels from The Cancer Proteome Atlas (TCPA) database[42,43] revealed that AMPK pT172 did not show correlation between HPRT1 expression or impact on prognosis of brain tumor patients (Supplementary Fig. 12l, m). These results implied that HPRT1/AMPK/RRM1 signal cascade was not activated in primary brain tumors. To understand whether these observations were due that these clinical

samples were obtained in prior to TMZ treatment, we collected another 50 recurrent GBM samples from patients who received TMZ treatment (Supplementary Data 3). High levels of HPRT1 expression, AMPK pT172, and RRM1 pT52 were correlated with poor survival expectation of these patients (Fig. 8d–f). Intriguingly, positive correlation was observed among HPRT1 expression, AMPK pT172, and RRM1 pT52 in these recurrent GBM samples (Fig. 8g–i), suggesting that TMZ strongly activated HPRT1/AMPK/RRM1 signal cascade in human GBM specimens and that this cascade may contribute to recurrence and refractory to TMZ of brain tumors.

## Discussion

TMZ treatment has been a standard care for GBM patients for more than 30 years but only very moderately extends patient survival time[44]. Understanding the mechanism underlying intrinsic TMZ chemoresistance is critical for improving treatment outcomes. In this report, we uncovered a critical bifunctional role of TMZ in GBM treatment. TMZ-derived methyldiazonium cations produce methylated purines, leading to DNA damage. However, TMZ-derived AICA counteracted methylated purine-mediated DNA damage by promoting DNA damage repair. Mechanistically, the TMZ metabolite AICA was converted by HPRT1 into AICAR, which activated AMPK and resulted in AMPK-mediated RRM1 phosphorylation at T52. This phosphorylation increased the binding affinity of ATP to RRM1, activated RNR, and subsequently increased the production of dNTPs for DNA damage repair. Treatment with 6-MP disrupted the "shield" by competitive inhibition of the binding of AICA to HPRT1 due to a comparably high affinity of 6-MP for HPRT1. Inhibition of HPRT1-mediated AICA catalysis by HPRT1 depletion or 6-MP treatment and expression of RRM1 T52A exacerbated TMZ-induced DNA damage by attenuating DNA damage repair, leading to increased cell apoptosis and sensitization of GBM to TMZ treatment in mice (Supplementary Fig. 13).

AMPK activation redirects metabolic pathways, including the activation of autophagy, to protect cells from harmful insults, such as chemotherapeutic drug treatment[17,45,46]. O⁶-methylguanine produced by the TMZ-derived methyldiazonium cations was reported to promote AMPK activation by enhancing ROS production[19]. However, elimination of methyldiazonium cations by diazo scavengers or reduction of ROS by NAC only moderately affected TMZ-induced AMPK activation. In contrast, blockade of TMZ-induced AICAR production substantially inhibited AMPK activation. In addition, mutation of the AMPK-phosphorylated RRM1 residue T52 to alanine strongly blunted DNA damage repair, indicating that TMZ-induced and HPRT1-mediated AICAR production is primarily responsible for TMZ-induced AMPK activation and that RRM1 T52 phosphorylation is critical for subsequent DNA damage repair.

AMPK has been suggested to be critical in DNA damage response. However, its role in regulating TMZ-induced DNA damage response and TMZ resistance is less clarified. A key mechanism for developing TMZ resistance is activating DNA damage repair systems, including MGMT, mismatch repair, base excision repair, and double-strand-break (DSB) repair consisting of non-homologous end joining (NHEJ) and homologous recombination (HR). In this study, we demonstrated that AMPK phosphorylates and activates RNR to promote dNTP production, which very likely promotes HR repair for TMZ resistance since that HR repair consumes large amount dNTPs. On the other hand, AMPK may promote TMZ resistance through activating NHEJ by phosphorylating and activating 53BP1[47]. Considering the different cellular localization of AMPK complex (cytoplasm) and 53BP1 (nucleus), this phosphorylating event may occur by a noncanonical population of nuclear-localized AMPK complex that specifically contains caspase-3-cleaved AMPK α1 subunit[48]. However, the involvement of (probably nuclear-localized) AMPK-mediated NHEJ pathway in TMZ resistance needs to be further validated. In addition, AMPK-mediated phosphorylation of exonuclease Exo1[49] may protect glioma cells from

TMZ-induced replication stress, thus contributing to TMZ resistance. Together, we propose that AMPK coordinates multi-layered mechanisms to promote TMZ resistance.

MGMT participates in a suicide reaction that specifically removes methyl moiety from the O-6-methylguanine adduct, resulting in irreversible inhibition of MGMT itself, as well as restoring guanine to its normal form without causing DNA breaks. However, MGMT proteins will run dry as TMZ outnumbers, inevitably leading to DNA damage. During the repair process, the supply of pooled nucleotides is critical for DNA damage repair and tumor cell survival. In this study, we demonstrate that HPRT1 mediates the production of AICAR from TMZ-derived AICA, which thereby activates AMPK to promote the nucleotide synthesis by phosphorylating and activating RRM1. This mechanism reveals the critical role of HPRT1 in getting GBM cells prepared ahead of TMZ-induced DNA damage. Therefore, we propose that the MGMT- and HPRT1-mediated repair processes are two layers of regulation of TMZ-induced DNA damage.

The drug 6-MP is currently being used for the treatment of ALL and CML. This metabolite is known to compete with the purine derivatives hypoxanthine and guanine for HPRT1 and is itself converted to thioinosine monophosphate (TIMP), which directly inhibits the de novo nucleotide synthesis pathway for purine ribonucleotide synthesis[50]. In addition to this established anticancer function, we found that 6-MP effectively blocks TMZ-derived AICAR production. Thus, our study revealed a "killing two birds with one stone" effect of 6-MP on GBM treatment. Importantly, we revealed that the TMZ metabolite AICA is a bona fide substrate of HPRT1 and is converted to the AMPK activator AICAR for tumor cell survival under TMZ treatment. The highly expressed HPRT1 in GBM specimens and the correlation of the expression levels of HPRT1 with the prognosis of TMZ-treated GBM patients highlight the pathological role of HPRT1-mediated metabolism of TMZ and RNR activation in TMZ chemoresistance. The synergetic effect on inhibiting brain tumor growth by combining 6-MP and TMZ treatment underscores the potential to overcome TMZ resistance and improve GBM treatment by combined administration of the currently clinically available drug 6-MP.

## Methods

### Ethical declarations

The animal study was approved by Nanjing Medical University Animal Experimental Ethics Committee. The use of clinical specimens was approved by the medical ethics committee of the First Affiliated Hospital of Nanjing Medical University.

### Materials

HRP-labeled anti-mouse (sc-525409) and anti-rabbit (sc-2357) secondary antibodies were purchased from Santa Cruz Biotechnology (Santa Cruz, CA, USA). Antibodies against ACC1 pS79 (11818), ACC1 (3676), AMPKα pT172 (50081), AMPKα (5831), AMPKα1 (4148), AMPK β1 (12063), γH2AX (9718), and α-Tubulin (3873) were purchased from Cell Signaling Technology (Beverly, MA, USA); Antibody against Flag (F3165) was purchased from Sigma-Aldrich (Shanghai, China); Antibodies against MGMT (ab108630) and UPRT (ab251653) were purchased from Abcam (Shanghai, China); Antibodies against APRT (21405-1-AP), OPRT (14830-1-AP), QPRT (25174-1-AP), HPRT1 (15059-1-AP), and RRM1 (10526-1-AP) were from Proteintech (Wuhan, China). Rabbit polyclonal antibodies against RRM1 T52 phosphorylation (RRM1 pT52) was produced by Affinity Biosciences LTD (Cincinnati, OH, USA). A peptide containing RRM1 T52 phosphorylation was injected into rabbits. The rabbit serum was collected and sequentially purified using an affinity column conjugated with non-phosphorylated and phosphorylated RRM1 T52 peptide, respectively, to exclude the antibodies recognizing non-T52- phosphorylated RRM1, followed by an affinity column conjugated with phosphorylated RRM1 T52 peptide to bind to and purify the RRM1 pT52 antibody. The antibody was then

eluted and concentrated. AICA (552410), Hypoxanthine (H9377), and Guanine (G11950) were from Sigma-Aldrich (Shanghai, China). PRPP (C5321), Mesna (A8469), and WR-1065 (A4487) were from APExBIO (Houston, USA). TMZ (S1237), NAC (S1623), A969662 (S2697), 6-MP (S1305), and O6-BG (S3658) were from Selleckchem (Boston, MA, USA). HisPur Ni-NTA Resin (88222) was from Thermo Scientific (Waltham, MA, USA). [γ-$^{32}$P]ATP (SRP-401) was from Hartmann Analytic.

## Mice

Luciferase-expressing GBM cells ($1 \times 10^4$ cells for MES28 and $1 \times 10^5$ cells for U87) were injected intracranially into 4-week-old female athymic old nude mice as previously described[51]. Briefly, mice were anesthetized with isoflurane. Cells were suspend in 5 μL of PBS, and intracranially injected using a 10 μL Hamilton syringe through a guide screw into the frontal lobe at a depth of 3 mm. Each group contains ten mice. Mice were fed with autoclaved food and water and maintained in s specific-pathogen-free facility with the housing conditions, $22 \pm 2$ °C, 12/12 light/dark cycle, $55 \pm 10\%$ humidity, and <400 lux. Tumor volumes were monitored by detecting the flux activity using bioluminescence imaging system at indicated times. The maximum permitted tumor burden was 50 mm³, which was not exceeded at any point. TMZ and 6-MP based treatment was initiated 7 days after the injection. The brain of each mouse was collected, fixed by 3.7% formaldehyde, and embedded in paraffin. The animal study was approved by Nanjing Medical University Animal Experimental Ethics Committee. Mice were euthanized and necropsied when exhibiting signs of declining neurologic status, performance status, or 20% loss of weight.

## Clinical samples

Primary and recurrent GBM clinical samples were obtained from the First Affiliated Hospital of Nanjing Medical University. Information on patients' sex was collected based on self-reporting. Patient group has been designed to be sex-homogeneous and sex-based analyses were not included in this study. Informed consent have been obtained from the participants. Paraffin-embedded GBM specimens were cut into 4 mm slices and stained with antibodies against HPRT1, AMPK pT172, and RRM1 pT52. The sample staining was quantitatively scored as previously described[51]. Briefly, the following proportion scores were assigned: 0 if 0% of the tumor cells were positively stained, 1 if 0%–1% of the tumor cells were positively stained, 2 if 2%–10% of the tumor cells were positively stained, 3 if 11%–30% of the tumor cells were positively stained, 4 if 31%–70% of the tumor cells were positively stained, and 5 if 71%–100% of the tumor cells were positively stained. Moreover, the staining intensity was rated on a scale of 0–3: 0, negative; 1, weak; 2, moderate; and 3, strong. The total score ranging from 0 to 8 was obtained by adding the proportion and intensity score, as shown in Supplementary Data 2 and 3. The use of clinical specimens was approved by the medical ethics committee of the First Affiliated Hospital of Nanjing Medical University.

## Cell culture

U87 (HTB-14), T98G (CRL-1690), and LN18 (CRL-2610) cells were obtained from ATCC. U251 (09063001) was purchased from Sigma-Aldrich (Shanghai, China). U251S, U251T3rd, N3S, and N3T3rd cells were constructed by our group as previously described[29]. Briefly, U251 and N3 cells were subcutaneously injected into nude mice. Mice were treated with three cycles of TMZ treatment. The TMZ-resistant cells were isolated from TMZ-treated xenografts, and the TMZ sensitive cells were isolated from vehicle-treated xenografts.

U87, U251, T98G, and LN18 cells were maintained in Dulbecco's modified Eagle's medium with 10% fetal bovine serum (Gibco). MES28 and GSC3028 cells were cultured in neurobasal medium with B27, 1-glutamine, sodium pyruvate, 10 ng/mL basic fibroblast growth factor, and 10 ng/mL epidermal growth factor as previously described[52,53]. For generation of HPRT1-, RRM1-, or MGMT-depleted stable cell lines, cells

were transfected with HPRT1, RRM1, or MGMT shRNA plasmids separately targeting HPRT1 and selected by puromycin.

All cells were confirmed to be negative for mycoplasma by PCR as described in the manuscript after every freeze-thaw cycle and before injection into mice. All cell lines were authenticated by PCR-single-locus-technology (Promega, USA. PowerPlex 21 PCR) analysis in "BMR Genomics s.r.l." (Italy).

## DNA construction and mutagenesis

Polymerase chain reaction (PCR)-amplified human HPRT1, RRM1, and MGMT were cloned into the pcDNA3.1/hygro(+)-Flag or pCold I vector. HPRT1 D138N, HPRT1 K166A, RRM1 T52A, and MGMT C145S were generated using the QuikChange site-directed mutagenesis kit (Stratagene, La Jolla, CA). shRNA-resistant (r) HPRT1 or RRM1 was constructed by introducing nonsense mutations in shRNA-targeting sites as previously described[51]. Briefly, hairpin constructs containing shRNAs targeting HPRT1 (5'-CCAGGTTATGACCTTGATTTA-3'), RRM1 (5'-CCCACAACTTTCTAGCTGTTT-3'), and MGMT (5'-GCTGTATTAAA GGAAGTGGCA-3') were synthesized and inserted into BamHI and MluI restriction sites of the pGIPZ vector following the manufacturer's construction.

## Transfection

Prior to transfection, cells at 60% confluence were plated in a 6-well plate. Before transfection, the culture medium was changed to fresh medium. The plasmids and Polyjet transfection reagent were mixed in FBS-free medium for 10 min. The mixture was subsequently added to the plates.

## Analysis of TMZ metabolites by HPLC-MS analysis

For determination of the levels of AICA, AICAR, $^{15}$N-AICA and $^{15}$N-AICAR in cells treated with TMZ or AICA, cellular extracts were prepared and analyzed by HPLC-MS. Briefly, U87 and MES28 cells were seeded in 10 cm dishes in triplicate. Cells were treated with TMZ or $^{15}$N-TMZ for 24 h. Iced PBS was used to wash the cells 4 times. The cells were collected using −40 °C iced methanol. The collected cells were then fast frozen in liquid nitrogen for 5 min and unfrozen at room temperature. Cellular extracts were centrifuged at 4000 rpm for 5 min at −4 °C. The supernatant was collected and centrifuged at $15,000 \times g$ for 1 min at −4 °C. The supernatant was injected into an Acquity UPLC BEH C18 SPE column and eluted using a mixture of $H_2O$ and MeOH (1:1). Then, eluted samples were transferred to new tubes followed by drying using nitrogen. Samples were resolved in 0.2% ammonium hydroxide in ammonium acetate. A 10 μL sample was injected into the Thermo Scientific Vanquish liquid chromatography (LC) system containing an ACQUITY UPLC BEH Amide Column (130 Å, 1.7 μm, 2.1 × 100 mm, 1/pk, Waters, Ireland). With a flow rate of 0.3 mL/min, the gradient elution program was as follows: 0 min (95% acetonitrile) – 3.0 min (95% acetonitrile) – 10.0 min (20% acetonitrile) – 15.0 min (20% acetonitrile) – 15.01 min (95% acetonitrile) – 17.0 min (95% acetonitrile). A Q Exactive Mass Spectrometer was used with electrospray ionization in both positive and negative modes to identify masses of candidate DBPs. Before each sequence of measurements, mass calibrations and mass accuracy were assessed to confirm that the resolution was always greater than 140,000 (at $m/z$ of 200), and the accuracy was kept within ±10 ppm. High-purity nitrogen was used to deliver sheath gas and aux gas at flow rates of 40 and 20 (arbitrary units), respectively. The spray voltage was set at 4 kV in positive polarity mode and 3 kV in negative polarity mode. The capillary temperature was set at 40 °C, and the source temperature was maintained at 8 °C. XCalibur v3.1.66.10 (Thermo, Waltham, MA) software was used for analysis.

External standard method was used for AICA and AICAR quantification. Briefly, the calibration curves of AICA and AICAR were lineated. The correlation coefficient $R^2$ of the regression equations exceeded the value 0.99 (Supplementary Data 4). Cellular AICA and

AICAR levels were calculated according to the calibration curves. For recovery measurement, new calibration curves of AICA and AICAR were lineated, and 1000, 10,000, 16,000, 30,000 ppb AICA and AICAR samples were prepared the same as cellular samples. Recoveries can be calculated based on theoretical values divided by the concentration of the SPE-treated samples (Supplementary Data 4).

## Purification of recombinant proteins

Expression of His-HPRT1, His-RRM1, His-RRM2, and His-MGMT was induced in bacteria, and protein purification was performed as previously described[51]. Briefly, 6x His-tagged recombinant proteins were cultured in 250 mL of lysogeny broth (LB) medium until the OD reached 0.6. Then, the bacteria were treated with 0.5 mM isopropyl β-D-1-thiogalactopyranoside (IPTG) overnight at 16 °C. The cell lysates were loaded onto a Ni-NTA column, washed with five column volumes of 20 mM imidazole, and eluted with 250 mM imidazole. The purified proteins were desalted using 10-kDa cut-through spin columns by washing with PBS.

## Detection of HPRT1-mediated AICAR production

The HPRT1 activity assay was conducted as previously described[54]. Briefly, active HPRT1 protein (100 ng) was incubated with its substrates (10 mM PRPP and 10 mM AICA) in 1 mL of kinase buffer (12 mM Tris-HCl, pH 7.3, 12 mM $MgCl_2$) at 37 °C for 2 h. The reaction was stopped by ice-cold water. Next, the reaction product was filtered through a C18 column, and the formation of AICAR was detected by HPLC-MS.

## RNR activity assay

The RNR activity assay was performed as previously described[55]. Briefly, 1 μg RRM1, 2 μg RRM2, 300 mM NaCl, 0.5 mM TCEP, and 50 mM HEPS, pH 7.5, were added together to form the mixture. Then, a 3.67 μL mixture was added to 46.33 μL reaction solution (0.8 mM CDP, dATP or ATP at the indicated concentration, 10 mM magnesium acetate, 10 mM DTT, and 20 mM Tris-HCl, pH 7.5) at 37 °C for 20 min. Ice-cold water was used to slow the reaction to nondetectable levels. The samples were filtered through a C18 column before HPLC analysis. The analysis was performed using a mobile phase of solution A (23%), solution B (57%), and solution A (20%). Solution A contained acetonitrile (7%), 23 g/L $KH_2PO_4$, pH 6.2, and KOH; Solution B contained acetonitrile (7%); and Solution C contained methanol (7%) and 3.56 g/L tetrabutylammonium bromide. The flow rate was kept at 0.4 mL/min. The dNDPs that came out within 12 min were detected by HPLC/MS.

## *Km* determination of HPRT1

For measurement of the Km of AICA, HX, Gua and 6-MP for HPRT1, purified recombinant HPRT1 protein (100 ng) was incubated with different concentrations of AICA, Hx, and G for 5 min followed by HPLC-MS measurement of the AICAR, IMG, GMP and TIMP concentrations.

## Immunoprecipitation and immunoblotting analysis

Protein samples were extracted from transfected cells using lysis buffer (50 mM Tris-HCl, pH 7.5, 0.01% SDS, 1% Triton X-100, 150 mM NaCl, 1 mM dithiothreitol, 0.5 mM EDTA, 100 μM PMSF, 100 μM leupeptin, 1 μM aprotinin, 100 μM sodium orthovanadate, 100 μM sodium pyrophosphate, and 1 mM sodium fluoride). Cellular extracts were then centrifuged at $12,000 \times g$ at 4 °C, and the supernatants were collected. For immunoprecipitation, 1 mg protein lysates were incubated with 10 μg antibodies overnight at 4 °C. Next, protein G-tagged beads were added to the lysates for another 3 h. Lysis buffer was used to wash the immunocomplexes 3 times. Immunoprecipitation products were analyzed using an immunoblotting assay against the indicated antibodies as previously described[51]. Briefly, immunoprecipitation products were separated on SDS-polyacrylamide gels and transferred onto polyvinylidene difluoride (PVDF) membranes. After blocking with 5% skim powdered milk for 2 h, the membranes were incubated with primary antibodies overnight at 4 °C and with HRP-conjugated secondary antibodies for 2 h at room temperature. The blots were visualized by ECL chemiluminescent reagent. Antibodies was diluted at 1:1000 for immunoprecipitation and 1:50 for immunoblotting analysis. The antibodies used in this study are listed in the Materials section.

## In vitro AMPK phosphorylation assay

Activate AMPK protein (Catalog #P47-10H) was purchased from SignalChem Biotech. The kinase activity of AMPK was validated using SAMS peptide as a substrate as described before[27]. In brief, AMPK was mixed with SAMS peptide (200 mM) in a solution containing 30 mM HEPES pH 7.4, 0.65 mM dithiothreitol, 0.02% Brij-35, 10 mM MgAc and 0.2 mM AMP. The reaction was started by the addition of 0.1 mM ATP (containing [γ-$^{32}$P]ATP at 1,000 c.p.m. pmol$^{-1}$), and was stopped after 20 min by adding 5 μL of 3% phosphoric acid, and 15 μL of the reaction mix were transferred to a piece of P81 phosphocellulose Whatman paper and washed extensively with phosphoric acid solution. Then, the paper was dried with acetone and radioactivity was counted by Cherenkov counting.

For in vitro kinase assays performed with "hot" [γ-$^{32}$P]ATP, active AMPK (10 ng) was incubated with purified WT His-RRM1 and His-RRM1 T52A (100 ng) in 25 μL of kinase buffer (50 mM Tris-HCl, pH 7.5, 100 mM KCl, 5 mM $MgCl_2$, 1 mM $Na_3VO_4$, 50 mM DTT, 5% glycerol, 0.2 mM AMP, 0.5 mM ATP, and 10 mCi [γ-$^{32}$P]ATP) at 25 °C for 30 min. The reactions were subjected to SDS-PAGE and autoradiography.

For in vitro kinase assays performed with "cold" ATP, active AMPK (SignalChem Biotech, Catalog #P47-10H) was incubated with purified WT His-RRM1 and His-RRM1 T52A (100 ng) in 25 μL of kinase buffer (50 mM Tris-HCl, pH 7.5, 100 mM KCl, 5 mM $MgCl_2$, 1 mM $Na_3VO_4$, 50 mM DTT, 5% glycerol, 0.2 mM AMP, and 0.5 mM ATP) at 25 °C for 30 min. The reactions were subjected to SDS-PAGE and then immunoblot analyses with corresponding antibodies.

The stoichiometry measurement of RRM1 phosphorylated by AMPK was performed as we have described previously[56]. 25 ng of active AMPK proteins (including AMPK α1/β1/γ1) were incubated with 200 ng of purified His-RRM1 in 25 μL of kinase buffer (50 mM Tris-HCl [pH 7.5], 100 mM KCl, 50 mM $MgCl_2$, 1 mM $Na_3VO_4$, 1 mM DTT, 5% glycerol). The reaction was started by addition of 0.1 mM [γ-$^{32}$P]ATP (250 cpm/pmol) at 25 °C. At various time points, samples were taken for SDS-PAGE analysis. Bands corresponding to RRM1 were cut directly from the gel and dissolved in vials containing 500 μL of 3% (w/w) $H_2O_2$ by heating 2 h at 80 °C. $^{32}$P incorporation was quantified by Cerenkov counting in a Beckman LS6500 scintillation counter and converted to moles of Pi incorporated per mole of RRM1.

## CRISPR/Cas9-mediated genomic editing

AMPKα1/2 double knockout (DKO) U87 cells were constructed using a CRISPR/Cas9 system as previously described[57]. Single-guided RNAs (sgRNAs) targeting AMPK α1 and α2 were designed using the CRISPR/Cas9 design tool (http://crispr.mit.edu/). The annealed guide RNA oligonucleotides were inserted into the PX458 vector (Addgene, Cambridge, MA) digested with the BbsI restriction enzyme. U87 cells at 60% confluence were seeded overnight followed by sgRNA (0.5 μg) transfection. Twenty-four hours after transfection, GFP-positive U87 cells were sorted by fluorescence-activated cell sorting (FACS) and seeded in 96-well plates. Genomic DNA was extracted from each colony, followed by sequencing of the PCR products spanning the target regions. The primers used for sgRNA cloning were as following, AMPKα1-F: 5′-caccgCTGGTGTGGATTATTGTCAC-3′; AMPKα1-R: 5′- aaa cGACCACACCTAATAACAGTGc-3′; AMPKα2-F: 5′-caccgACGTTATTTAA GAAGATCCG-3′; AMPKα2-R: 5′-aaacTGCAATAAATTCTTCTAGGCc-3′.

## Proliferation assay

A total of 1000 U87 or MES28 suspended in 200 μL medium were plated in a 96-well plate. CellTiter-Glo (Promega, Madison, WI, USA)

was used to measure cell proliferation according to the manufacturer's instructions. All data were normalized to those of day 1.

## IC50 measurement

GBM cells were exposed to increased concentrations of TMZ or 6-MP. Cell viability was measured at 24 h after treatment. The IC50 of TMZ for GBM cells was calculated using GraphPad software.

## Quantification of O6-mG in xenograft tissues

Methyl group of O6-mG can be transferred to cysteine 145 (C145) of MGMT. When mutating this residue into serine, the resultant MGMT C145S mutant is still able to recognize O6-mG, but fails to remove methyl group of O6-mG, resulting in its retention on the DNA regions containing O6-mG. Based on this, we purified MGMT C145S mutant tagged with Flag as a "sticky" probe for detecting O6mG levels in xenograft tissues. After tumor cell implantation, the brain was isolated and sequentially dehydrated with 15% and 30% sucrose, respectively, before embedding in OCT and frozen on dry ice. OCT molds were sectioned at 10 μm thickness. Sections were washed with PBS and incubated in 0.1 mM of Flag-MGMT C145S recombinant protein for 2 h at 37 °C, followed by incubation with formaldehyde for 5 min. Sections were then washed with PBS, incubated in PBS with 3% bovine serum albumin (BSA) at room temperature for 1 h and incubated overnight at 4 °C in primary antibody against Flag. The following day, sections were washed three times in PBS, incubated with a secondary antibody against the appropriate species (1:500) diluted in PBS with 1.5% BSA at room temperature for 1 h, and washed three times in PBS. Cell nuclei were stained using DAPI and fluorescence images were captured using a ZEISS AxioScan7 microscope.

## Quantification of O6-mG in cells by gold nanoprobes

To quantify O6-mG levels in GBM cells, we synthesized phosphoramidites of ExBenzi nucleoside as previously described[18]. 1H-naphtho[2,3-d]imidazol-2(3H)-one (1.103 g, 5.500 mmol), anhydrous THF (50 mL), and NaH (60% in mineral oil, 0.333 g, 8.250 mmol) was mixed and flushed with $N_2$, and re-suspended with chunky gold precipitate at 24 °C. Then, 1-(α)-Chloro-3,5-di-O-ptoluoyl-2-deoxy-D-ribose (2.140 g, 5.500 mmol) was added dropwise followed by stirring for 1.5 h. The reaction was quenched with $H_2O$ (15 mL) and concentrated to dryness with rotatory evaporation. Next, the solid was suspended in 1:1 DCM/MeOH (50 mL) and centrifuged to remove the supernatant. The precipitation was re-suspended by 20 mL MeOH and centrifuged again. The supernatants were pooled and concentrated, then purified via flash chromatography using a 0-10% MeOH in DCM gradient. A portion of fully deprotected nucleoside (65 mg, 0.218 mmol) was obtained. Remaining bis-toluoyl protected (44 mg, 0.082 mmol) and mono-toluoyl protected (120 mg, 0.306 mmol) were combined and dissolved in THF (9 mL), to which NaH (60% in mineral oil, 29 mg, 0.730 mmol) was added, followed by MeOH (1 mL). After 2 min, phosphoramidites of ExBenzi nucleoside were dried by rotatory evaporation and purified via flash chromatography using a linear 3–10% MeOH in DCM gradient. Two types of nanoprobes were prepared: the detection nanoprobe that was functionalized with a 5′-thiolmodified oligonucleotide (sequence: 5′- HS - (T)10 CCT ACG–3′), and the discriminating nanoprobe that was functionalized with a 5′-ExBenzi (3′-thiol_ExBenzi_1). U87 cells were pretreated with or without 10 μM Mesna or 40 μM WR-1065, followed by TMZ treatment for 30 min. Genomic DNA extracted from U87 cells was fragmented by ultrasonication. ExBenzi nanoprobes (1 nM) and detection probes (1 nM) were mixed with fragmented genomic DNA (100 ng). The mixture was heated to 70 °C for 10 min, and then allowed to cool to 25 °C. The mixture were then incubated for 6 h at room temperature until full aggregation had occurred. Upon target-induced aggregation, absorbance ratios (A700/A530) were measured.

## Apoptosis analysis

A total of $1 \times 10^5$ U87 and MES28 cells pretreated with or without TMZ were collected and washed twice with ice-cold PBS. Then, 100 μL of 1x binding buffer was used to suspend the cells. Cells were stained with 5 μL of Annexin V-FITC (Catalog: A211; Vazyme, Nanjing, China) and 5 μL of PI staining solution at room temperature for 10 min. Apoptosis was analyzed by flow cytometry and quantified using BD FACSDiva software as previously described[58]. Gating strategy for Annexin V-FITC was provided (Supplementary Fig. 14).

## Terminal deoxynucleotidyl transferase dUTP nick end labeling (TUNEL) analysis

U87- and MES28-derived tumors were cut into 4 mm slices. The rate of apoptotic cells in tumors was analyzed using the TUNEL BrightGreen Apoptosis Detection Kit (Vazyme) according to the manufacturer's instructions.

## Quantification and statistical analysis

Statistical analyses were conducted with a two-tailed unpaired Student's $t$-test unless specifically indicated. All data represent the mean ± standard deviation (SD) of six independent experiments/samples unless otherwise specified. Differences in means were considered statistically significant at $P < 0.05$. Significance levels are: $*P < 0.05$; $**P < 0.001$. Analyses were performed using the Prism software.

## Reporting summary

Further information on research design is available in the Nature Portfolio Reporting Summary linked to this article.

## Data availability

The data that support the conclusions of this study are provided in the Article, Supplementary, and Source Data files, and also available from the corresponding authors. Source data are provided in this paper. Raw data have been deposited at https://www.scidb.cn/s/RRFrE3. Source data are provided in this paper.

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

## Acknowledgements

We thank Dr. Chaojun Li from the State Key Laboratory of Reproductive Medicine and Offspring Health of Nanjing Medical University for his advice on this study. This work was supported by the National Natural Science Foundation of China 82272651 (Y.Y.), 82072765 (X.Q.), 82030074 and 82188102 (Z.L.), 81972610 (Z.S.), 81974389 (Y.Y.), 82172667 (X.W.), and 82002914 (X.G.), Jiangsu Province Capability Improvement Project through Science, Technology and Education (ZDXK202225, Y.Y.) and Postgraduate Research & Practice Innovation Program of Jiangsu Province (J.Y.). Z.L. is the Kuancheng Wang Distinguished Chair.

## Author contributions

X.Q., Z.L., and Y.Y. conceived and designed the study and interpreted the results. J.Y., X.F.W., X.G., F.D., Z.S., and Z.G. performed most of the experiments; G.H. and N.Z. performed the HPLC and mass spectrometry. D.C. provided support on MGMT analyses. J.Z., Y.C., and J.J. provided support on clinical analyses. S.A. and X.X.W. provided glioblastoma stem cells and technical support. F.L., Q.W., and Q.Z. provided advice and reagents. X.Q. wrote the manuscript with comments from all authors.

## Competing interests

The authors declare no competing interests.

## Additional information

[1]Department of Neurosurgery, The First Affiliated Hospital of Nanjing Medical University, 210029 Nanjing, China. [2]Institute for Brain Tumors, Collaborative Innovation Center for Cancer Personalized Medicine, and Center for Global Health, Nanjing Medical University, 211166 Nanjing, China. [3]Department of Nutrition and Food Hygiene, School of Public Health, Nanjing Medical University, 210029 Nanjing, China. [4]Department of Health Inspection and Quarantine, School of Public Health, Nanjing Medical University, 211166 Nanjing, China. [5]China Exposomics Institute, 200120 Shanghai, China. [6]Affiliated Hospital of Nanjing University of Chinese Medicine, 210029 Nanjing, China. [7]Department of Medicinal Chemistry, School of Pharmacy, Nanjing Medical University, 211166 Nanjing, China. [8]Department of Neurological Surgery, UPMC Children's Hospital of Pittsburgh, Pittsburgh, PA 15224, USA. [9]Department of Radiation Oncology, The First Affiliated Hospital of Nanjing Medical University, 210029 Nanjing, China. [10]Department of Cell Biology, School of Basic Medical Sciences, Nanjing Medical University, 211166 Nanjing, China. [11]Department of Bioinformatics, Nanjing Medical University, 211166 Nanjing, China. [12]Department of Clinical Pharmacology, School of Pharmacy, Nanjing Medical University, 211166 Nanjing, China. [13]National Health Commission Key Laboratory of Antibody Technologies, Nanjing Medical University, 211166 Nanjing, China. [14]Zhejiang Provincial Key Laboratory of Pancreatic Disease, The First Affiliated Hospital, Zhejiang University School of Medicine, 310029 Hangzhou, China. [15]Institute of Translational Medicine, Zhejiang University Cancer Center, Zhejiang University, 310029 Hangzhou, China. [16]Key Laboratory of Modern Toxicology of Ministry of Education, School of Public Health, Nanjing Medical University, 211166 Nanjing, China. [17]Present address: Gusu School, Nanjing Medical University, 215006 Suzhou, China. [18]These authors contributed equally: Jianxing Yin, Xiefeng Wang, Xin Ge. ✉e-mail: YYPL9@njmu.edu.cn; zhiminlu@zju.edu.cn; xqianmedres@njmu.edu.cn

