## [Peer Review File · Nature Communications]

Reviewers' Comments:

Reviewer #1:

Remarks to the Author:

In this study Yin et al. identify a paradoxical effect of TMZ and identify HPRT1 as a key regulator of the efficacy of TMZ in GBM. They propose that HPRT1 catalyzes the conversion of AICA to AICAR. AICAR in turn activates an AMPK—RRM1 cascade leading to increased production of dNTPs and increased DNA repair, counteracting the cytotoxic effects of TMZ. The findings of this study are quite interesting and potentially important since TMZ is the only chemotherapeutic drug that has even a modest effect on survival in GBM patients. Another strength is that the authors uncover a detailed mechanism to support their model. The data derived from the experimental models are generally quite convincing and rigorous, and support the conclusions of this paper. The study could be strengthened and the clinical relevance enhanced by validating some of the findings in TMZ treated GBM tissue and providing additional evidence for a role for this pathway in TMZ resistance. I have some suggestions that may help to strengthen the paper as outlined below.

1. The authors should discuss the MGMT status of the cell lines used in the study. They should also examine and comment on whether the biological impact of HPRT1-DNA repair cascade triggered by TMZ is influenced by MGMT status of the tumor.

2. Is it possible to detect activation of this cascade in TMZ treated GBM tissue by IHC especially in the high HPRT1 patients in their dataset or perhaps using TCGA data? For example is it possible to detect phosphorylation of AMPK and or phosphorylation of RRM1. Does pAMPK from the TCGA/TCGA database show an impact on prognosis?

3. The authors should provide some clarification of the GBM tissue used. In most clinical settings, the tumor is resected first and then the patient gets radiation and TMZ. It is not clear whether the 100 GBMs examined are post-TMZ treatment or removed prior to TMZ, or a combination of both. Obviously enhanced activity of this pathway in post TMZ treatment recurrent GBM tissue would be quite compelling.

4. If this TMZ DNA repair pathway plays an important role in TMZ resistance, is there evidence that this pathway is more active in TMZ resistant vs. TMZ sensitive experimental models (that cannot be attributed to MGMT status) ? is this pathway likely to play a role in primary resistance to TMZ or secondary resistance . Or both?

5. A minor concern is that the differences between the various groups in the bioluminescence data shown in Fig 5A are not very convincing. Generally, bioluminescence—although quite useful for detecting tumor formation-- may not be the optimal method for detecting tumor size. However, the survival analysis shown for each model is quite convincing.

Reviewer #2:

Remarks to the Author:

This is an interesting paper on the mechanism of resistance of brain tumor cells to TMZ, a chemotherapeutic agent, and a proposed TMZ/6-MP combination therapy approach to overcome that resistance. The study was quite comprehensive, employing an array of pharmacological and genetic approaches to manipulate key metabolites or targets and include endpoints in metabolite levels, DNA repair activity, gene expression, enzyme activity, and cell survival in in vitro experiments with cultured tumor cells, and tumor and survival outcomes in mouse models. The data support the main conclusion that TMZ degradation product AICA is converted to AICAR in tumor cells by HPRT1, and that AICAR stimulated RNR, leading to increased DNA repair activities. The effects of co-treatment with 6-MP also looked convincing.

A few major concerns should be addressed:

1. A key part of the data is on AICA and AICAR level determinations, which was done using LC-MS. However, there was no description of how that was done. The only details on mass spectrometry are for metabolite identification, not for quantification. In the present form, it is unclear whether those quantitative data are robust or not, which weakens the main conclusions. Details including analytical methods, internal standards, recovery, and sensitivity should all be reported.

2. The conclusion that the diazonium ion did not contribute to the simulation of DNA repair is a bit weak. Though two approaches were used to remove the diazonium ions, there was no indication that those approaches are sufficient to remove the diazonium ion under the conditions used. Validation of the effectiveness of those approaches would help.

3. While 6-MP treatment was effective on tumor cells, it is important to address whether the same treatment also affected DNA repair in normal cells, which could potentially increase levels of O6mG DNA adduct there and cause greater risk of generating new tumors in other parts of the body. This can be tested by measuring O6-mG levels in tumors and non-tumor tissues in mice exposed to the combination therapy, compared to controls or those exposed to monotherapy.

Some minor points to consider:

1. line 37: "less unclarified" should be "less clarified"
2. line 101: "upstream metabolite" should be "precursor"

Reviewer #3:

Remarks to the Author:

This paper puts forward an interesting argument as to why the drug temozolomide (TMZ, used to treat glioblastoma) is less effective than it otherwise might be. TMZ is metabolized within cells not only to methyl diazonium ions, which cause DNA damage, but also to AICA, which is converted to AICAR (also known as ZMP) by the enzyme HPRT1. ZMP is an AMP analogue that was already known to activate AMP-activated protein kinase (AMPK), which then promotes DNA repair and hence enhances survival of the cells, thus reducing the efficacy of TMZ. The authors go on to provide evidence, using mouse xenografts, that 6-mercaptopurine (6MP), a drug already used clinically that competitively inhibits HPRT1, enhances the therapeutic effects of TMZ. They also provide evidence from analysis of human cancer databases that this mechanism might be relevant to human glioblastoma.

Although their results are in general quite convincing, one relatively weak part of the paper is the evidence that AMPK activates ribonucleotide reductase via direct phosphorylation of its catalytic subunit, RRM1 (see points 6 and 7 below).

MAJOR POINTS:

1. General: although the authors suggest a novel mechanism by which AMPK may promote DNA repair (phosphorylation and activation of ribonucleotide reductase), it is worth pointing out that at least two other mechanisms by which this may occur have been previously reported, i.e. phosphorylation of EXO1 (DOI: 10.1016/j.molcel.2019.04.003) and phosphorylation of 53BP1 (DOI: 10.1016/j.celrep.2021.108713). Also relevant to this, it has been reported that AMPK complexes containing the alpha-1 isoform accumulate in the cell nucleus during DNA damage, due to cleavage of alpha-1 by caspase-3 (DOI:10.1016/j.celrep.2022.110761).

2. Line 66: they talk here about "intrinsic and acquired chemoresistance", but it is not clear how the mechanisms they propose would be relevant to acquired chemoresistance.

3. Line 98 and Fig. 2: can the authors provide any reassurance that the concentrations of TMZ (and indeed AICA) used in these and subsequent cell culture experiments (which seem very high - up to 0.4-0.5 mM) are relevant to plasma concentrations of these compounds in humans treated with TMZ?

4. Lines 133-134: as well as the K_m for AICA being 5-10-fold higher than those for guanine or hypoxanthine, the V_{max} also appears to be about 10-fold lower, which the authors do not comment upon. While this does not rule out their hypothesis, it does suggest that AICA is a rather poor substrate for HPRT1.

5. Lines 151-152 and Supp. Fig. 4a, b: the symbols at the bottom are not aligned properly with the columns, nor are the statistics bars at the top. This makes it very difficult to interpret these results.

6. Line 165: Scansite might give an indication that Thr-52 is an AMPK site, but in fact it is a rather poor fit to the AMPK consensus motif. Although it does have leucine at -5 and +4 with respect to the phosphoamino acid, essentially all recognized AMPK targets also have basic residues at -4 and/or -3 [see DOI:10.1016/j.tcb.2015.10.013], which is not the case with this site. Note also that serine is much preferred over threonine as the amino acid phosphorylated by AMPK.

7. Line 167: while the evidence that AMPK is required for phosphorylation of Thr52 on RRM1 in intact cells (Fig. 4f) is good, in the light of point (6) above there remains some doubt that phosphorylation of RRM1 at Thr52 is directly mediated by AMPK. The crucial experiment here is Fig. 4d, but I could not find what the "activated AMPK" mentioned in the Figure legend and Methods section actually was. What subunit isoforms were present, how was it "activated" and what kinase activity (measured using another substrate, such as the SAMS peptide) was used in these assays? Without this information, it is impossible to judge how good an AMPK substrate RRM1 really is. Note also that a positive signal obtained using a phosphospecific antibody does not prove that the degree of phosphorylation of the site is significant, which may require experiments using 32P- or 33P-labelled ATP.

MINOR POINTS:

8. Line 79: the full name of AMPK is "AMP-activated protein kinase", not "AMP-dependent protein kinase" – note that it is not completely dependent upon the presence of AMP.

9. Line 127: "which originally..." seems inappropriate here – how about "whose primary role is to catalyze...."?

10. Line 173 and Fig. 4F: what are C1 and C2 - are these two clones of KO cells?

11. Lines 183-184: this reviewer is not an expert on ribonucleotide reductase, but as far as I am aware ATP is a regulator, not a substrate. The term "Km... for ATP" therefore seems inappropriate.

12. Lines 206 and Supp. Fig. 5b: it is not clear from the information provided what bioluminescence marker is being used.

13. Lines 206 and following, and Fig. 5: there is far too much information in Fig. 5, which is more or less illegible when printed on standard sized paper (A4 or US letter).

14. Line 237: if the estimate of Km is approximate, why give it to four figures?

15. Lines 457 and 463: here they are describing HPRT1 and RNR assays: they are not kinase assays.

REVIEWER COMMENTS

Reviewer #1 - GBM therapy (Remarks to the Author):

In this study Yin et al. identify a paradoxical effect of TMZ and identify HPRT1 as a key regulator of the efficacy of TMZ in GBM. They propose that HPRT1 catalyzes the conversion of AICA to AICAR. AICAR in turn activates an AMPK—RRM1 cascade leading to increased production of dNTPs and increased DNA repair, counteracting the cytotoxic effects of TMZ. The findings of this study are quite interesting and potentially important since TMZ is the only chemotherapeutic drug that has even a modest effect on survival in GBM patients. Another strength is that the authors uncover a detailed mechanism to support their model. The data derived from the experimental models are generally quite convincing and rigorous, and support the conclusions of this paper. The study could be strengthened and the clinical relevance enhanced by validating some of the findings in TMZ treated GBM tissue and providing additional evidence for a role for this pathway in TMZ resistance. I have some suggestions that may help to strengthen the paper as outlined below.

Response: We greatly appreciate the reviewer's acknowledgement of the potential significance of this study and the insightful comments, which are essential for improvement of this manuscript.

1. The authors should discuss the MGMT status of the cell lines used in the study. They should also examine and comment on whether the biological impact of HPRT1-DNA repair cascade triggered by TMZ is influenced by MGMT status of the tumor.

Response: We included several cell lines in which U87 and U251 cells are MGMT-null, whereas LN229, LN18, MES28, and GSC3028 are MGMT-intact (Supplementary Fig. 6a). Depletion of HPRT1 had no discriminated effects on the repair dynamic of TMZ-induced DNA damage when comparing between MGMT-null U87 and MGMT-intact MES28 cells (Supplementary Fig. 5f). In consistent with these observations, depletion of MGMT in MGMT-intact LN229 and LN18 cells or overexpression of MGMT in MGMT-null U87 and U251 cells had no additive effects on HPRT1 depletion-induced DNA repair or apoptosis in response to TMZ treatment (Supplementary Fig. 6b-e). These results suggested that HPRT1-mediated DNA repair is independent of MGMT status.

MGMT participates in a suicide reaction that specifically removes methyl moiety from the O⁶-methylguanine adduct, resulting in irreversible inhibition of MGMT itself, as well as restoring guanine to its normal form without causing DNA breaks. However, MGMT proteins will run dry as TMZ outnumbers, inevitably leading to DNA damage. During the repair process, the supply of pooled nucleotides is critical for DNA damage repair and tumor cell survival. In this study, we demonstrate that HPRT1 mediates the production of AICAR from TMZ-derived AICA, which thereby activates

AMPK to promote the nucleotide synthesis by phosphorylating and activating RRM1. This mechanism reveals the critical role of HPRT1 in getting GBM cells prepared ahead of TMZ-induced DNA damage. Therefore, we propose that the MGMT- and HPRT1-mediated repair processes are two layers of regulation of TMZ-induced DNA damage. We have included this in the Discussion section.

2. Is it possible to detect activation of this cascade in TMZ treated GBM tissue by IHC especially in the high HPRT1 patients in their dataset or perhaps using TCGA data? For example is it possible to detect phosphorylation of AMPK and or phosphorylation of RRM1. Does pAMPK from the TCGA/TCGA database show an impact on prognosis?

Response: We examined HPRT1/AMPK/RRM1 signal cascade in our 100 GBM tissues. The levels of HPRT1 expression showed no correlation with AMPK pT172 or RRM1 pT52 (Supplementary Fig. 12f, g). The levels of AMPK pT172 demonstrated positive correlation with that of RRM1 pT52 in all these samples (n = 100, Supplementary Fig. 12h), as well as in those samples expressing with low or high levels of HPRT1 (n = 50, respectively, Supplementary Fig. 12i). In addition, levels of AMPK pT172 or RRM1 pT52 did not show any correlation with survival of these patients (Supplementary Fig. 12j, k).

RRM1 pT52 was just newly identified in this study. No data about it was available in TCGA database. Therefore, we only checked AMPK pT172 in this database. No correlation was found between levels of HPRT1 expression and AMPK pT172 in brain tumors (Supplementary Fig. 12l). In addition, no impact of AMPK pT172 on prognosis of either low grade gliomas or GBMs was found in this database (Supplementary Fig. 12m).

As pointed out in Question #3, the 100 GBM samples we collected and those samples included in TCGA/TCGA database were primary tumors prior to TMZ treatment. In the circumstances without TMZ stimulation, HPRT1 would not be a upstream that activates AMPK/RRM1 pT52 cascade for GBM survival.

3. The authors should provide some clarification of the GBM tissue used. In most clinical settings, the tumor is resected first and then the patient gets radiation and TMZ. It is not clear whether the 100 GBMs examined are post-TMZ treatment or removed prior to TMZ, or a combination of both. Obviously enhanced activity of this pathway in post TMZ treatment recurrent GBM tissue would be quite compelling.

Response: Following this suggestion, we collected another 50 GBM samples that were recurrent and post-TMZ treatment. We have included the information on the previously 100 primary and the newly 50 recurrent samples in Supplementary Tables 3 and 4. Positive correlations were observed among

HPRT1 expression, AMPK pT172, and RRM1 pT52 (Fig. 7g-i). Intriguingly, high levels of HPRT1 expression, AMPK pT172, and RRM1 pT52 predict worse survival of recurrent GBM patients, respectively (Fig. 7d-f).

4. If this TMZ DNA repair pathway plays an important role in TMZ resistance, is there evidence that this pathway is more active in TMZ resistant vs. TMZ sensitive experimental models (that cannot be attributed to MGMT status) ? is this pathway likely to play a role in primary resistance to TMZ or secondary resistance . Or both?

Response: We employed our previously generated TMZ resistant cell lines and their paired sensitive counterparts, U251T3rd vs. U251S, and N3T3rd vs. N3S (Zeng et al., *Cancer Lett* 2018; PMID: 30102952). We confirmed the resistant capacity of these cells to TMZ (Supplementary Fig. 7a). HPRT1 expression levels were much increased in these TMZ-resistant cells (Supplementary Fig. 7b). In line with the increased HPRT1 expression levels, the TMZ-resistant cells exhibited much accelerated activation of AMPK and the subsequent phosphorylation of RRM1 T52 in response to both dosage- and time-dependent TMZ treatment (Supplementary Fig. 7c, d). Depletion of HPRT1 in TMZ-resistant cells suppressed TMZ-induced AMPK activation and phosphorylation of RRM T52 (Supplementary Fig. 7e), and subsequently increased the sensitivity to TMZ treatment as demonstrated by reduced IC50 to TMZ and increased apoptotic effects (Supplementary Fig. 7f, g). These data suggested that HPRT1-mediated AMPK/RRM1/DNA repair pathway plays a role in secondary resistance to TMZ.

Glioma stem cells (GSCs) are primarily resistant to TMZ (Chen et al., *Nature* 2012; PMID: 22854781). In accordance, we observed that GSCs exhibited much higher IC50 to TMZ when compared to differentiated glioma cell lines (Supplementary Fig. 5k), and that the IC50 to TMZ was positively correlated with HPRT1 expression levels in these cells (Supplementary Fig. 5j, k). Furthermore, HPRT1 was demonstrated to be critical for AMPK activation, RRM1 T52 phosphorylation, and DNA damage repair in response to TMZ treatment in GSC cell MES28 (Fig. 3e, 4h, and Supplementary Fig. 5f). These data suggested that HPRT1-mediated AMPK/RRM1/DNA repair pathway plays a role in primary resistance to TMZ. Together, we propose that HPRT1-mediated DNA damage repair pathway plays critical roles in both primary and secondary resistance to TMZ.

5. A minor concern is that the differences between the various groups in the bioluminescence data shown in Fig 5A are not very convincing. Generally, bioluminescence—although quite useful for detecting tumor formation-- may not be the optimal method for detecting tumor size. However, the survival analysis shown for each model is quite convincing.

Response: We agree with the reviewer's comment. To make up the deficiency of bioluminescence, we stained the animal brain tumor slices with H&E and showed that the tumor sizes were in good accordance with bioluminescent signals (Supplementary Fig. 8c, 9b, 11f).

Reviewer #2 - AMPK (Remarks to the Author):

This is an interesting paper on the mechanism of resistance of brain tumor cells to TMZ, a chemotherapeutic agent, and a proposed TMZ/6-MP combination therapy approach to overcome that resistance. The study was quite comprehensive, employing an array of pharmacological and genetic approaches to manipulate key metabolites or targets and include endpoints in metabolite levels, DNA repair activity, gene expression, enzyme activity, and cell survival in *in vitro* experiments with cultured tumor cells, and tumor and survival outcomes in mouse models. The data support the main conclusion that TMZ degradation product AICA is converted to AICAR in tumor cells by HPRT1, and that AICAR stimulated RNR, leading to increased DNA repair activities. The effects of co-treatment with 6-MP also looked convincing.

Response: We greatly appreciate the reviewer's acknowledgement of the potential significance of this study and the insightful comments, which are essential for improvement of this manuscript.

A few major concerns should be addressed:

1. A key part of the data is on AICA and AICAR level determinations, which was done using LC-MS. However, there was no description of how that was done. The only details on mass spectrometry are for metabolite identification, not for quantification. In the present form, it is unclear whether those quantitative data are robust or not, which weakens the main conclusions. Details including analytical methods, internal standards, recovery, and sensitivity should all be reported.

Response: We updated the detailed quantification information for AICA, AICAR, ¹⁵N-AICA, and ¹⁵N-AICAR in Methods section. External standard method was used for AICA/AICAR quantification, where both the correlation coefficient R^2 of the regression equations for AICA and AICAR exceeded the value 0.99 (Supplementary Table 5). In addition, the recovery rates were more than 60% for both metabolites (Supplementary Table 5). The sensitivity of LC-MS for detecting AICA and AICAR was evaluated by measuring both metabolites in samples containing decreasing concentrations of AICA and AICAR. As shown in Supplementary Table 5, AICA and AICAR could be effectively detected even when AICA and AICAR are in low concentration, indicating the highly detective sensitivity.

2. The conclusion that the diazonium ion did not contribute to the simulation of DNA repair is a bit weak. Though two approaches were used to remove the diazonium ions, there was no indication that those approaches are sufficient to remove the diazonium ion under the conditions used. Validation of the effectiveness of those approaches would help.

Response: We determined the effectiveness of Mesna and WR-1065 on consuming diazonium ions through both *in vitro* and *in vivo* assays. In an *in vitro* assay, we incubated serial dosages of TMZ-

derived metabolites with fixed concentration of Mesna or WR-1065 for 5 min. HPLC analyses revealed that the production of methylated Mesna or WR-1065 (Methyl-Mesna or Methyl-WR-1065) was stoichiometrically correlated with TMZ dosages (Supplementary Fig. 2b), suggesting effective removal of diazonium ion by Mesna and WR-1065 *in vitro*. To further validate the effectiveness of Mesna and WR-1065 on removing TMZ-derived diazonium ions *in vivo*, we synthesized gold nanoparticles that specifically recognize O6-mG, the product of diazonium ion on DNA, according to the reported study (Trantakis et al., *J Am Chem Soc* 2016; PMID: 27314828). These nanoparticles effectively detected O6-mG when treating MGMT-null U87 cells with TMZ for 30 min (Supplementary Fig. 2c). Intriguingly, pretreating these cells with Mesna or WR-1065 greatly reduced the levels of O6-mG (Supplementary Fig. 2c), suggesting effective removal of diazonium ions.

3. While 6-MP treatment was effective on tumor cells, it is important to address whether the same treatment also affected DNA repair in normal cells, which could potentially increase levels of O6mG DNA adduct there and cause greater risk of generating new tumors in other parts of the body. This can be tested by measuring O6-mG levels in tumors and non-tumor tissues in mice exposed to the combination therapy, compared to controls or those exposed to monotherapy.

Response: Measuring O6-mG levels in animal tissues is challenging since that the synthesized gold nanoparticles (Question #2) can only detect O6-mG in cultured cells instead of xenograft tumor tissues, and that no commercial antibody recognizing O6-mG is available. To solve this problem, we developed an immunofluorescent assay-based method (refer to Method section). Methyl group of O6-mG can be transferred to cysteine 145 (C145) of MGMT. When mutating this residue into serine, the resultant MGMT C145S mutant is still able to recognize O6-mG, but fails to remove methyl group of O6-mG, resulting in its retention on the DNA regions containing O6-mG. Inspired by this, we purified Flag-MGMT C145S to recognize O6-mG in the xenograft samples. This MGMT-C145S/O6-mG recognition was further fixed by formaldehyde. Finally, the levels of O6-mG can be determined by immunofluorescent staining with Flag (Supplementary Fig. 11l). O6-benzylguanine (O6-BG), a guanine analog that is also recognized by MGMT C145S, competitively blocked the interaction between MGMT C145S with O6-mG and subsequently reduced the immunofluorescent intensity of Flag (Supplementary Fig. 11l), further supporting the specificity of this method.

Through this newly developed assay, we observed that the combination of 6-MP with TMZ caused profound O6-mG in xenograft tumors when comparing with TMZ treatment alone (Supplementary Fig. 11m). However, we did not observe additive effects of 6-MP/TMZ combination on the levels of O6-mG than TMZ treatment alone in the tumor-adjacent (Supplementary Fig. 11n) or normal brain tissues (Supplementary Fig. 11o). These observations may be due to the expression of MGMT which

effectively removes O6-mG in normal brain tissues, therefore protecting normal brain tissues from damaging by 6-MP/TMZ combination treatment.

Some minor points to consider:

1. line 37: “less unclarified” should be “less clarified”
2. line 101: “upstream metabolite” should be “precursor”

Response: These points have been corrected.

Reviewer #3 - AMPK and AICAR (Remarks to the Author):

This paper puts forward an interesting argument as to why the drug temozolomide (TMZ, used to treat glioblastoma) is less effective than it otherwise might be. TMZ is metabolized within cells not only to methylidiazonium ions, which cause DNA damage, but also to AICA, which is converted to AICAR (also known as ZMP) by the enzyme HPRT1. ZMP is an AMP analogue that was already known to activate AMP-activated protein kinase (AMPK), which then promotes DNA repair and hence enhances survival of the cells, thus reducing the efficacy of TMZ. The authors go on to provide evidence, using mouse xenografts, that 6-mercaptopurine (6MP), a drug already used clinically that competitively inhibits HPRT1, enhances the therapeutic effects of TMZ. They also provide evidence from analysis of human cancer databases that this mechanism might be relevant to human glioblastoma. Although their results are in general quite convincing, one relatively weak part of the paper is the evidence that AMPK activates ribonucleotide reductase via direct phosphorylation of its catalytic subunit, RRM1 (see points 6 and 7 below).

Response: We greatly appreciate the reviewer's acknowledgement of the potential significance of this study and the insightful comments, which are essential for improvement of this manuscript.

MAJOR POINTS:

1. General: although the authors suggest a novel mechanism by which AMPK may promote DNA repair (phosphorylation and activation of ribonucleotide reductase), it is worth pointing out that at least two other mechanisms by which this may occur have been previously reported, i.e. phosphorylation of EXO1 (DOI: 10.1016/j.molcel.2019.04.003) and phosphorylation of 53BP1 (DOI: 10.1016/j.celrep.2021.108713). Also relevant to this, it has been reported that AMPK complexes containing the alpha-1 isoform accumulate in the cell nucleus during DNA damage, due to cleavage of alpha-1 by caspase-3 (DOI:10.1016/j.celrep.2022.110761).

Response: Several studies have suggested that AMPK plays important roles in DNA damage response. However, its role in regulating TMZ-induced DNA damage response and TMZ resistance is less clarified. A key mechanism for developing TMZ resistance is activating DNA damage repair systems, including O-6-methylguanine DNA methyltransferase (MGMT) that removes TMZ-induced DNA adducts, mismatch repair, base excision repair, and double-strand-break (DSB) repair consisting of non-homologous end joining (NHEJ) and homologous recombination (HR). In this study, we demonstrated that AMPK phosphorylates and activates RNR to promote dNTP production, which very likely promotes HR repair for TMZ resistance since that HR repair consumes large amount dNTPs. On the other hand, AMPK may promote TMZ resistance through activating NHEJ by phosphorylating and activating 53BP1 (Jiang et al., *Cell Rep* 2021; PMID: 33596428). Considering the different cellular

localization of AMPK complex (plasma) and 53BP1 (nucleus), this phosphorylating event may occur by a noncanonical population of nuclear-localized AMPK complex that specifically contains caspase-3-cleaved AMPK α 1 subunit (Cheratta et al., *Cell Rep* 2021; PMID: 35508122). However, the involvement of (probably nuclear-localized) AMPK-mediated NHEJ pathway in TMZ resistance needs to be further validated. In addition, AMPK-mediated phosphorylation of exonuclease Exo1 (Li et al., *Mol Cell* 2019; PMID: 31053472) may protect glioma cells from TMZ-induced replication stress, thus contributing to TMZ resistance. Together, we propose that AMPK coordinates multi-layered mechanisms to promote TMZ resistance. We have put this part in the Discussion section.

2. Line 66: they talk here about “intrinsic and acquired chemoresistance”, but it is not clear how the mechanisms they propose would be relevant to acquired chemoresistance.

Response: The original intention of “intrinsic and acquired TMZ chemoresistance” statement in Introduction section was to introduce the mechanisms contributing to TMZ resistance.

Together this comment with Question #4 from Reviewer 1, we did additional experiments using our previously generated cell lines that acquired TMZ resistance and their paired sensitive counterparts, U251T3rd vs. U251S, and N3rd vs. N3S (Zeng et al., *Cancer Lett* 2018; PMID: 30102952). We confirmed the resistant capacity of these cells to TMZ (Supplementary Fig. 7a). HPRT1 expression levels were much increased in these TMZ-resistant cells (Supplementary Fig. 7b). In line with the increased HPRT1 expression levels, the TMZ-resistant cells exhibited much accelerated activation of AMPK and the subsequent phosphorylation of RRM1 T52 in response to both dosage- and time-dependent TMZ treatment (Supplementary Fig. 7c, d). Depletion of HPRT1 in TMZ-resistant cells suppressed TMZ-induced AMPK activation and phosphorylation of RRM T52 (Supplementary Fig. 7e), and subsequently increased the sensitivity to TMZ treatment as demonstrated by reduced IC₅₀ to TMZ and increased apoptotic effects (Supplementary Fig. 7f). These data suggested that HPRT1-mediated AMPK/RRM1/DNA repair pathway plays a role in acquired/secondary resistance to TMZ.

We also tested this mechanisms in glioma stem cells (GSCs) that are intrinsically/primarily resistant to TMZ (Chen et al., *Nature* 2012; PMID: 22854781). In accordance, we observed that GSCs exhibited much higher IC₅₀ to TMZ when compared to differentiated glioma cell lines, and that the IC₅₀ to TMZ was positively correlated with HPRT1 expression levels in these cells (Supplementary Fig. 5j, k). Furthermore, HPRT1 was demonstrated to be critical for AMPK activation, RRM1 T52 phosphorylation, and DNA damage repair in response to TMZ treatment in GSC cell MES28 (Fig. 3e, 4h, and Supplementary Fig. 5f). These data suggested that HPRT1-mediated AMPK/RRM1/DNA

repair pathway plays a role in primary resistance to TMZ. Together, we propose that HPRT1-mediated DNA damage repair pathway plays critical roles in both intrinsic and acquired resistance to TMZ.

3. Line 98 and Fig. 2: can the authors provide any reassurance that the concentrations of TMZ (and indeed AICA) used in these and subsequent cell culture experiments (which seem very high - up to 0.4-0.5 mM) are relevant to plasma concentrations of these compounds in humans treated with TMZ?

Response: The standard-of-care Stupp protocol for GBM patients recommends 75-200 mg of TMZ per square meter of body-surface area per day. TMZ penetrates blood-brain barrier well, making it possible that brain tumor cells receive sufficient concentration of plasma TMZ. According to a previous publication (Ostermann et al., *Clin Cancer Res* 2004; PMID: 15173079), the max TMZ levels were 13.99 $\mu\text{g/ml}$ in the plasma. Assuming an adult with 5 liter of blood volume, the calculated plasma concentration of TMZ is 0.36 mM. These data support the rationale of treating GBM cells with such a relative high concentration of TMZ.

4. Lines 133-134: as well as the K_m for AICA being 5-10-fold higher than those for guanine or hypoxanthine, the V_{max} also appears to be about 10-fold lower, which the authors do not comment upon. While this does not rule out their hypothesis, it does suggest that AICA is a rather poor substrate for HPRT1.

Response: We agree with the comment that AICA is a poor substrate for HPRT1. The V_{max} of HPRT1 for AICA was 7.18 ± 0.38 nmol/min (Fig. 3d), which was comparable to that of hypoxanthine (8.81 ± 0.18 nmol/min) and guanine (9.51 ± 0.31 nmol/min) (Supplementary Fig. 3f). We are sorry that we mislabeled the y-axis of IMP and GMP production in the original Supplementary Fig. 3f, which misled the reviewer's interpretation about V_{max} .

Considering that AICA is derived from TMZ and therefore not a natively existing cellular metabolite, the HPRT1-mediated AICA conversion does not necessarily take place in cells. However, in the specific circumstance such as treating cells with TMZ, the intracellular concentration of AICA (Supplementary Fig. 3g) greatly exceeds the K_m of HPRT1 for AICA, which compensates the low efficiency of reaction and warrants sufficient production of AICAR to activate AMPK. We have included the V_{max} values and comments in the main text.

5. Lines 151-152 and Supp. Fig. 4a, b: the symbols at the bottom are not aligned properly with the columns, nor are the statistics bars at the top. This makes it very difficult to interpret these results.

Response: We are sorry for this negligence. The labeling was properly aligned (Supplementary Fig. 4a, b).

6. Line 165: Scansite might give an indication that Thr-52 is an AMPK site, but in fact it is a rather poor fit to the AMPK consensus motif. Although it does have leucine at -5 and +4 with respect to the phosphoamino acid, essentially all recognized AMPK targets also have basic residues at -4 and/or -3 [see DOI:10.1016/j.tcb.2015.10.013], which is not the case with this site. Note also that serine is much preferred over threonine as the amino acid phosphorylated by AMPK.

Response: We agree that T52 of RRM1 does not fit AMPK consensus well. In addition to its well-fit substrate ACC1 at S79, AMPK has been reported to phosphorylate several non-canonical substrates such as histone H2B at S36 (Bungard et al., *Science* 2010; PMID: 20647423) and phosphoribosyl pyrophosphate synthetase 1 (PRPS1) at S180 (Qian et al., *Cancer Discov* 2018; PMID: 29074724). In addition to serines as phosphorylated target amino acids of AMPK, those non-canonical substrates with threonines phosphorylated by AMPK were also reported, such as 6-pyruvoyltetrahydropterin synthase (PTPS) at T58 (Zhao et al., *Mol Cell* 2020; PMID: 31628042) and FOXO1 at T649 (Saline et al., *J Biol Chem* 2019; PMID: 31308176). These information suggest the existence of broader substrate spectrums of AMPK in addition to those substrates with matched consensus. We have included this information in Supplementary Fig. 4g.

In addition to prediction by fitness of its consensus motif, experimental validation is quite critical for confirming an AMPK substrate. Our *in vitro* and *in vivo* experiments combining with RRM1 T52A mutant and generated antibody against RRM1 pT52 collectively supported that AMPK phosphorylates RRM1 at T52 (Fig. 4d-h), albeit with the possibility that this phosphorylation reaction is less efficient than that of AMPK for its substrates with matched consensus.

7. Line 167: while the evidence that AMPK is required for phosphorylation of Thr52 on RRM1 in intact cells (Fig. 4f) is good, in the light of point (6) above there remains some doubt that phosphorylation of RRM1 at Thr52 is directly mediated by AMPK. The crucial experiment here is Fig. 4d, but I could not find what the “activated AMPK” mentioned in the Figure legend and Methods section actually was. What subunit isoforms were present, how was it “activated” and what kinase activity (measured using another substrate, such as the SAMS peptide) was used in these assays? Without this information, it is impossible to judge how good an AMPK substrate RRM1 really is. Note also that a positive signal obtained using a phosphospecific antibody does not prove that the degree of phosphorylation of the site is significant, which may require experiments using ³²P- or ³³P-labelled ATP.

Response: The active AMPK consisting of $\alpha 1/\beta 1/\gamma 1$ subunits was purchased from SignalChem Biotech (Catalog #P47-10H). This information has been included in Methods section. Following the reviewer's suggestion, we validated that the purchased AMPK was active through measuring its activity on SAMS peptide (Supplementary Fig. 4h).

We performed the suggested "hot" assay by incubating active AMPK with bacterially purified WT His-RRM1 or His-RRM1 T52A phosphorylation-dead mutant in the presence of $[\gamma\text{-}^{32}\text{P}]\text{ATP}$, followed by autoradiography. WT RRM1, but not RRM1 T52A mutant, was phosphorylated by AMPK (Fig. 4d), further confirming that AMPK phosphorylates RRM1 at T52.

MINOR POINTS:

8. Line 79: the full name of AMPK is "AMP-activated protein kinase", not "AMP-dependent protein kinase" – note that it is not completely dependent upon the presence of AMP.

Response: We corrected this typo.

9. Line 127: "which originally..." seems inappropriate here – how about "whose primary role is to catalyze...."?

Response: The suggested rephrase is much better. We corrected it.

10. Line 173 and Fig. 4F: what are C1 and C2 - are these two clones of KO cells?

Response: Yes, C1 and C2 refer to clones of AMPK $\alpha 1/2$ KO cells. We have included these information in the figure legends of main text and supplementary text.

11. Lines 183-184: this reviewer is not an expert on ribonucleotide reductase, but as far as I am aware ATP is a regulator, not a substrate. The term "Km... for ATP" therefore seems inappropriate.

Response: We greatly thank the reviewer for pointing this mistake out. The parameter for ATP in RNR activity is Kd, while for dATP is IC50. We have corrected this in the main text.

12. Lines 206 and Supp. Fig. 5b: it is not clear from the information provided what bioluminescence marker is being used.

Response: The bioluminescence signal is luciferase. The GBM cells injected into mice brain were stably expressing luciferase. We added this information in the animal study parts in both Results and Methods sections.

13. Lines 206 and following, and Fig. 5: there is far too much information in Fig. 5, which is more or less illegible when printed on standard sized paper (A4 or US letter).

Response: We reorganized both Fig 5 and Fig 6 to make the information much concise.

14. Line 237: if the estimate of K_m is approximate, why give it to four figures?

Response: We corrected the K_m value from 43.36 nM to 43.36 ± 5.74 nM, and deleted “approximately” in the corresponding main text.

15. Lines 457 and 463: here they are describing HPRT1 and RNR assays: they are not kinase assays.

Response: We split this part into “HPRT1 activity assay” and “RNR activity assay”, respectively.

Reviewers' Comments:

Reviewer #1:

Remarks to the Author:

The authors have adequately addressed all of the concerns raised in my original review and I am pleased to recommend publication of this interesting work.

Reviewer #2:

Remarks to the Author:

I am happy with the revisions.

Reviewer #3:

Remarks to the Author:

The authors have made genuine attempts to answer my comments on version 1. A few relatively minor points remain:

1. Discussion, line 398: I think that the localization of the AMPK complex should be "(cytoplasm)" not "(plasma)".

2. Figure 4d: they have added an experiment using ³²P-labelled ATP, which shows that the T52A mutation eliminates phosphorylation of RRM1. This is useful, but it should also have been possible to calculate the stoichiometry of phosphorylation of T52 (which ideally should at least approach 1 mole/mole), based on estimates of the molar quantity of RRM1, the amount of radioactivity in the RRM1 band, and the known specific radioactivity of the labelled ATP. They should consider doing this.

3. Lines 287-288: they have deleted the word "approximately" from the original version, but I suspect that it is not possible to estimate K_m to a precision of four significant figures. I suggest that " 43.36 ± 5.74 nM" should be given as " 44 ± 6 nM" and " 12.92 ± 2.48 μ M" as " 13 ± 2 ".

REVIEWERS' COMMENTS

Reviewer #1 (Remarks to the Author):

The authors have adequately addressed all of the concerns raised in my original review and I am pleased to recommend publication of this interesting work.

Response: We greatly appreciate the reviewer's insightful comments which greatly improved the significance of this manuscript.

Reviewer #2 (Remarks to the Author):

I am happy with the revisions.

Response: We greatly appreciate the reviewer's insightful comments which greatly improved the significance of this manuscript.

Reviewer #3 (Remarks to the Author):

The authors have made genuine attempts to answer my comments on version 1. A few relatively minor points remain:

Response: We greatly appreciate the reviewer's insightful comments which greatly improved the significance of this manuscript.

1. Discussion, line 398: I think that the localization of the AMPK complex should be "(cytoplasm)" not "(plasma)".

Response: We replaced "plasma" with "cytoplasm". Thank you very much for your careful verbatim reading.

2. Figure 4d: they have added an experiment using ³²P-labelled ATP, which shows that the T52A mutation eliminates phosphorylation of RRM1. This is useful, but it should also have been possible to calculate the stoichiometry of phosphorylation of T52 (which ideally should at least approach 1 mole/mole), based on estimates of the molar quantity of RRM1, the amount of radioactivity in the RRM1 band, and the known specific radioactivity of the labelled ATP. They should consider doing this.

Response: We did the suggested assay. The calculated stoichiometry of RRM1 phosphorylated by AMPK is ~1 mol of phosphate per mol of RRM1 proteins (Fig. 4f). This assay together with both *in vitro* and *in vivo* observations (Fig. 4d, 4e, 4h) pinpointed that AMPK phosphorylates RRM1 at T52.

3. Lines 287-288: they have deleted the word "approximately" from the original version, but I suspect that it is not possible to estimate K_m to a precision of four significant figures. I suggest that " 43.36 ± 5.74 nM" should be given as " 44 ± 6 nM" and " 12.92 ± 2.48 μ M" as " 13 ± 2 ".

Response: We corrected the K_m values as suggested.